# *Gaga*: Group Any Gaussians via 3D-aware Memory Bank

**Weijie Lyu**                                                      *wlyu3@ucmerced.edu*
*University of California, Merced*

**Xueting Li**                                                      *xuetingl@nvidia.com*
*NVIDIA Research*

**Abhijit Kundu**                                                   *abhijitkundu@google.com*
*Google DeepMind*

**Yi-Hsuan Tsai**                                                   *yhtsai@atmanity.io*
*Atmanity Inc.*

**Ming-Hsuan Yang**                                                 *myang37@ucmerced.edu*
*University of California, Merced*

**Reviewed on OpenReview:** *https://openreview.net/forum?id=cC1TLyK3iW*

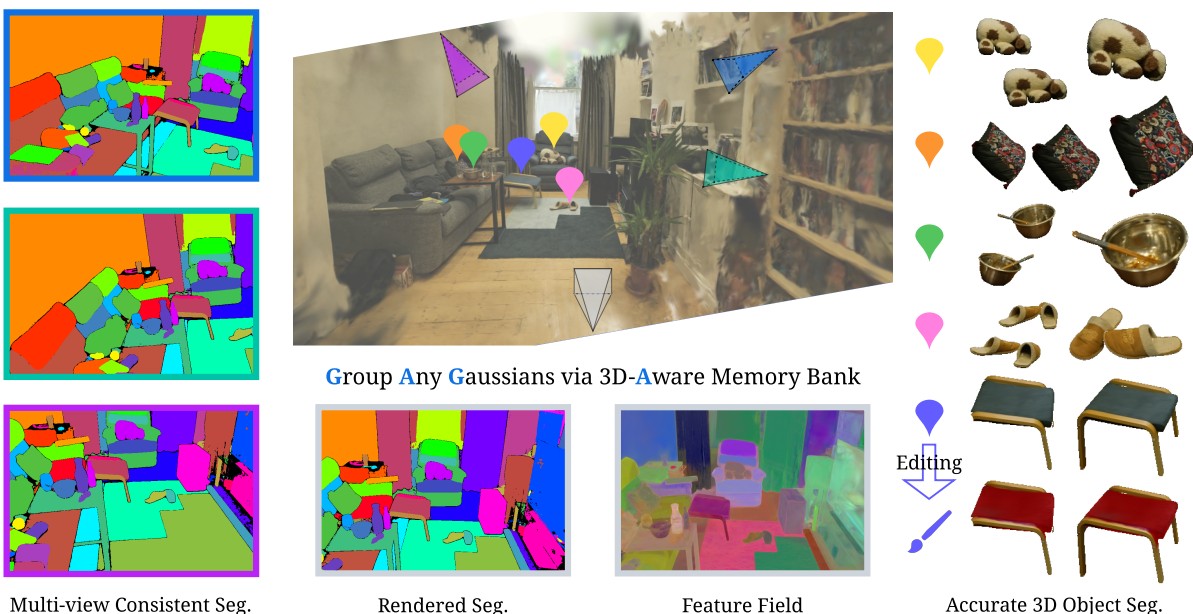

Group Any Gaussians via 3D-Aware Memory Bank

| Multi-view Consistent Seg. | Rendered Seg. | Feature Field | Accurate 3D Object Seg. |

Figure 1: ***Gaga* groups any Gaussians** in an open-world 3D scene and renders multi-view consistent segmentation (pixels of the same region across views are represented with the same color). By employing a 3D-aware memory bank, we eliminate the label inconsistency that exists in 2D segmentation predicted by foundational models and assign each mask across different views a universal group ID. This enables the process of lifting 2D segmentation to a consistent 3D segmentation. *Gaga* produces accurate 3D object segmentation, achieving high-quality results for downstream applications such as scene manipulation (*e.g.* changing the cushion's color of the footstool to maroon). Project page: https://wlyu.me/Gaga.

## Abstract

We introduce *Gaga*, a framework that reconstructs and segments open-world 3D scenes by leveraging inconsistent 2D masks predicted by zero-shot class-agnostic segmentation models. Contrasted to prior 3D scene segmentation approaches that rely on video object tracking or contrastive learning methods, *Gaga* utilizes spatial information and effectively associates object masks across diverse camera poses through a novel 3D-aware memory bank. By

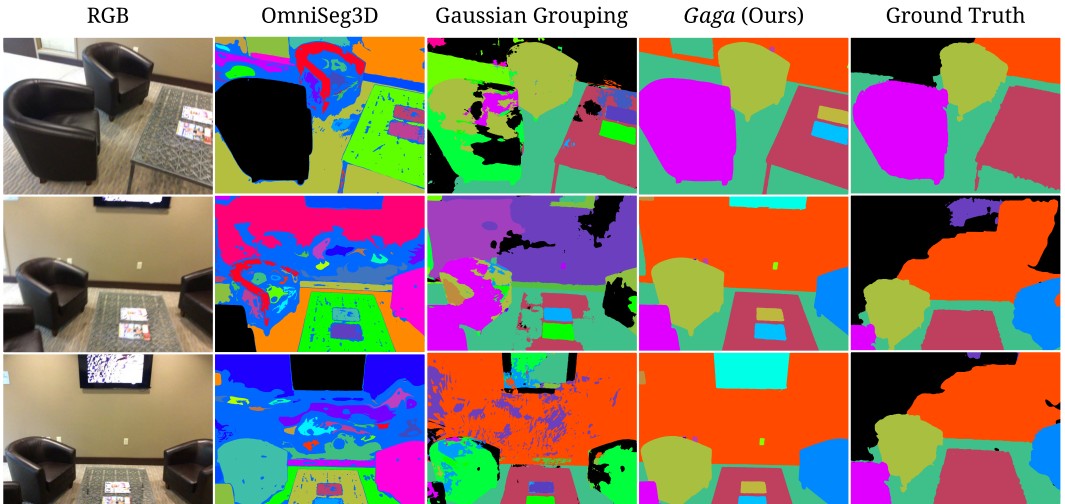

| RGB | OmniSeg3D | Gaussian Grouping | *Gaga* (Ours) | Ground Truth |

Figure 2: **Comparison of Rendered Segmentation.** Contrastive learning-based methods, such as OmniSeg (Ying et al., 2024), do not provide unique mask labels to each segmentation group, leading to inconsistencies across multiple views (*e.g.*, the coffee table). Gaussian Grouping (Ye et al., 2024) addresses multi-view segmentation by utilizing a video tracker, but it often misidentifies objects when similar items are present (*e.g.*, the leather sofa) and struggles with significant camera perspective changes. In contrast, *Gaga* ensures multi-view consistent segmentation masks, overcoming these limitations.

eliminating the assumption of continuous view changes in training images, *Gaga* demonstrates robustness to variations in camera poses, particularly beneficial for sparsely sampled images, ensuring precise mask label consistency. Furthermore, *Gaga* accommodates 2D segmentation masks from diverse sources and demonstrates robust performance with different open-world zero-shot class-agnostic segmentation models, significantly enhancing its versatility. Extensive qualitative and quantitative evaluations demonstrate that *Gaga* performs favorably against state-of-the-art methods, emphasizing its potential for real-world applications such as 3D scene understanding and manipulation.

# 1    Introduction

Effective open-world 3D segmentation is essential for scene understanding and manipulation. Despite notable advancements in 2D open-world segmentation techniques, exemplified by Segment Anything (SAM) (Kirillov et al., 2023) and EntitySeg (Qi et al., 2023), extending these methodologies to the realm of 3D encounters the challenge of ensuring consistent mask label assignment across multi-view images. Specifically, masks of the same object across different views may have different mask label IDs, as the 2D class-agnostic segmentation model independently processes the multi-view images. Naively lifting these inconsistent masks to 3D introduces ambiguity and leads to inferior results in 3D scene segmentation. Existing works (Ying et al., 2024; Kim et al., 2024; Ye et al., 2024; Dou et al., 2024; Choi et al., 2024) address this issue either by contrastive learning or video object tracking. One line of work builds upon 3D scenes represented as 3D Gaussians (Kerbl et al., 2023). In this approach, a feature vector is learned for each 3D Gaussian through a contrastive learning paradigm. The method encourages similar features for 3D Gaussians whose projections fall within the same segmentation mask, while pushing apart those belonging to different segmentation regions. The segmentation maps for each view are then obtained by clustering these feature vectors. However, these methods do not assign a specific mask label to each 3D segmentation group, resulting in inconsistent multi-view segmentation, as shown in Fig. 2. Alternatively, some methods (Ye et al., 2024; Dou et al., 2024) tackle multi-view inconsistency in 3D scene segmentation by treating multi-view images as a video sequence and adopting an off-the-shelf video object tracking method (Cheng et al., 2023a). Nevertheless, this design assumes minimal view changes between multiview images, a condition that may not always hold in real-world

3D scenes, particularly when input views are sparse. Consequently, these methods often struggle with similar objects or occluded objects that intermittently disappear and reappear in the sequence, as shown in Fig. 2.

We identify a fundamental limitation in existing open-world 3D scene segmentation methods that leads to 3D inconsistency: the inadequate exploitation of 3D information inherently provided by the scene. Our main intuition is that masks of the same object across different views shall correspond to the same group of 3D Gaussians. Hence, we can assign identical universal mask IDs to masks from different views when there is a large overlap between their corresponding 3D Gaussian groups.

Based on this intuition, we propose *Gaga*, which groups any 3D Gaussians and renders consistent 3D class-agnostic segmentation across different views. Given a collection of posed RGB images, we first employ Gaussian Splatting (Kerbl et al., 2023) to reconstruct the 3D scene and extract 2D masks using an open-world segmentation method (Kirillov et al., 2023; Qi et al., 2023). Subsequently, we iteratively build a 3D-aware memory bank that collects and stores the Gaussians grouped by category. For each input view, we project each 2D mask into a 3D space using camera parameters and search the memory bank for the category with the largest overlap with the unprojected mask. Based on the degree of overlap, we assign the mask to an existing category or create a new one. Finally, following the mask association process described above, we can get a set of multi-view consistent 2D segmentation masks. Finally, we learn a feature vector (*i.e.*, identity encoding (Ye et al., 2024)) for each 3D Gaussian that encodes its category information. Specifically, we splat the feature vectors onto the 2D image and decode a segmentation map through a linear layer. The predicted masks are then compared with the segmentation masks obtained from our 3D-aware memory bank for supervision. This approach ensures that our identity encoding remains multi-view consistent, thanks to the consistent segmentation masks across different viewpoints. The contributions of this work are:

- We propose a framework for reconstructing and segmenting 3D scenes from inconsistent 2D masks generated by open-world segmentation models.
- To resolve mask inconsistency across views, we design a 3D-aware memory bank that collects Gaussians of the same semantic group and uses them to align 2D masks across diverse views.
- Our method can leverage any 2D class-agnostic segmentation mask, making it readily applicable to novel-view image and mask synthesis.
- Experiments on diverse datasets and challenging scenarios, including sparse input views, demonstrate the qualitative and quantitative effectiveness of our method.

## 2 Related Work

**Segmenting and Tracking Anything in 2D.** Segment Anything (SAM) (Kirillov et al., 2023) and EntitySeg (Qi et al., 2023) demonstrate the effectiveness of large-scale training in image segmentation, thus establishing a pivotal foundation for open-world segmentation methods. Subsequent studies (Yang et al., 2023; Cheng et al., 2023b;a) further extend the applicability of SAM to video data by leveraging video object segmentation algorithms to propagate SAM masks. Conversely, acquiring data for training their 3D counterparts poses a challenge, given that existing large-scale 3D datasets with annotated segmentation (Straub et al., 2019; Dai et al., 2017) focus on indoor scenarios.

**NeRF-based 3D Segmentation.** Neural Radiance Fields (NeRFs) (Mildenhall et al., 2020) model scenes as continuous volumetric functions, learned through neural networks that map 3D coordinates to scene radiance. This approach facilitates the capture of intricate geometric details and the generation of photo-realistic renderings, offering novel view synthesis capabilities. Semantic-NeRF (Zhi et al., 2021) initiates the incorporation of semantic information into NeRFs and enables the generation of semantic masks for novel views. Note that semantic segmentation masks do not face the challenge of ambiguous mask ID across views. Numerous methods expand the scope by introducing instance modeling and matching instance masks relying on existing 3D bounding boxes (Liu et al., 2023c; Fu et al., 2022), resorting to the cost-based linear assignment during training (Siddiqui et al., 2023; Wang et al., 2023) or directly training instance-specific MLPs (Kundu et al., 2022). However, most of these methods are developed based on ground truth segmentation and tailored for scene modeling within specific domains. They often entail high computational costs and lack substantial evidence of their performance in open-world scenarios. Leveraging SAM's open-world

segmentation capability, SA3D (Cen et al., 2023b) endeavors to recover a 3D consistent mask by tracing 2D masks across adjacent views with user guidance. Similarly, Chen et al. (Chen et al., 2023) distill SAM encoder features into 3D and query the decoder. In contrast, *Gaga* achieves multi-view consistency without user intervention, offering segmentation for all objects rather than a single instance. Garfield (Kim et al., 2024) densely samples SAM masks and trains a scale-conditioned affinity field supervised on the scale of each mask deprojected to 3D, which focuses on 3D clustering rather than segmentation.

**Gaussian-based 3D Segmentation.** As an alternative to NeRF and its variants (Mildenhall et al., 2020; Chen et al., 2022; Müller et al., 2022; Tancik et al., 2023), Gaussian Splatting (Kerbl et al., 2023; Chen & Wang, 2024; Wu et al., 2024; Yu et al., 2024) has recently emerged as a powerful approach to reconstruct 3D scenes via real-time radiance field rendering. By representing the scene as 3D Gaussians from posed images, it achieves photorealistic novel view synthesis with high reconstruction quality and efficiency. Additionally, manipulating 3D Gaussians for scene editing is more straightforward compared to NeRF's representation. SAGA (Cen et al., 2023a) renders a 2D SAM feature map and uses a SAM guidance loss to learn 3D segmentation from the ambiguous 2D masks. Similar to (Cen et al., 2023b), this method requires user input and only provides segmentation for one object at a time. Feature 3DGS (Zhou et al., 2024) distills LSeg (Li et al., 2022) and SAM features to 3D Gaussians and decodes rendered features to obtain segmentation. However, it fails to provide consistent segmentation across views. Gaussian Grouping (Ye et al., 2024) and CoSSegGaussians (Dou et al., 2024) use a video object tracker (Cheng et al., 2023a) to associate masks across different views. However, in scenarios with significant changes in camera poses between frames, such approaches struggle to maintain accuracy. OmniSeg3D (Ying et al., 2024) and Click-Gaussians (Choi et al., 2024) use contrastive learning-based method to attach each Gaussian with a feature, while the features of two Gaussians are encouraged to be similar if their projections are within the same segmentation mask. However, such methods do not provide segmentation mask labels for each segmented Gaussian group. More recently, InstanceGaussian (Li et al., 2024) proposes a unified appearance-semantic 3D Gaussian representation with progressive joint training and bottom-up category-agnostic instance aggregation; in contrast, Gaga focuses on explicit geometry-based cross-view mask association through a 3D-aware memory bank built from off-the-shelf 2D masks.

## 3 Proposed Method

### 3.1 Preliminaries

**Gaussian Splatting.** Gaussian Splatting (Kerbl et al., 2023) has significantly advanced the 3D representation field by combining the benefits of implicit and explicit 3D representations. Specifically, a 3D scene is parameterized by a set of 3D Gaussians $\{G_i\}$. Each Gaussian $G_i = \{p_i, s_i, q_i, \alpha_i, c_i\}$ is defined by its position $p_i = \{x, y, z\} \in \mathbb{R}^3$, scale $s_i \in \mathbb{R}^3$, orientation $q \in \mathbb{R}^4$, opacity $\alpha \in \mathbb{R}$ and color features $c$ encoded by spherical harmonics (SH) coefficients. Gaussian Splatting employs the splatting pipeline, wherein 3D Gaussians are projected onto the 2D image space using the world-to-frame transformation matrix corresponding to each camera pose. Gaussians projected to the same coordinates $(x, y)$ (represented as $i \in N$) are blended in depth order and weighted by their opacity $\alpha$ to produce the color $c_{x,y}$ of each pixel:

$$c_{x,y} = \sum_{i \in N} c_i \alpha_i \prod_{j=1}^{i-1} (1 - \alpha_j). \tag{1}$$

### 3.2 Grouping Any Gaussians via 3D-aware Memory Bank

Given a set of posed images, we aim to reconstruct a 3D scene with semantic labels for segmentation rendering. To this end, we first leverage Gaussian Splatting for scene reconstruction. We then employ open-vocabulary 2D segmentation methods such as SAM (Kirillov et al., 2023) or EntitySeg (Qi et al., 2023) to predict class-agnostic segmentation for each input view. However, because the segmentation model processes each input view independently, the resulting masks are not naturally multi-view consistent. To resolve this issue, existing works (Ye et al., 2024; Dou et al., 2024) carry out a mask association process that tries to consolidate inconsistent segmentation masks from different views. For example, GaussianGrouping (Ye et al., 2024) and CoSSegGaussians (Dou et al., 2024) apply a video tracker to associate inconsistent 2D masks of

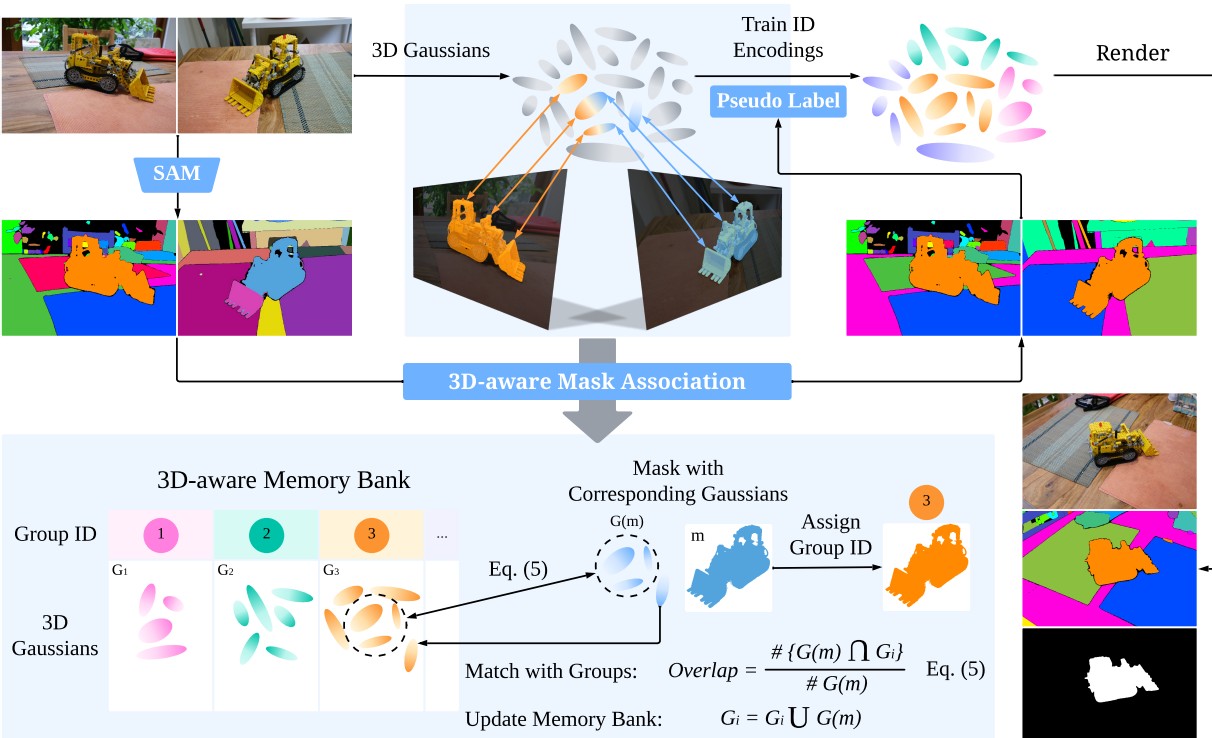

Figure 3: **Overview of *Gaga*.** *Gaga* reconstructs 3D scenes using Gaussian Splatting and adopts any open-world model to generate 2D segmentation masks. To eliminate the 2D mask label inconsistency, we design a mask association process, where a 3D-aware memory bank is employed to assign a consistent group ID across different views to each 2D mask based on the 3D Gaussians projected to that mask (Sec. 3.2). Specifically, we find the corresponding Gaussians projected to 2D mask and assign the mask with the group ID in the memory bank with the maximum overlapped Gaussians (Eq. 5) After 3D-aware mask association process, we use masks with multi-view consistent group IDs as pseudo labels to train an identity encoding on each 3D Gaussian for segmentation rendering.

different views, assuming similarity between nearby input views. This assumption, however, may not hold for all 3D scenes, particularly when the input views are sparse, as demonstrated in Fig. 2.

*Gaga* is inspired by the fundamental disparity between the task of mask association across multiple views and object tracking in a video: the latter does not exploit 3D information in the scene. To reliably generate consistent masks across different views, we propose a method that leverages 3D information without relying on any assumptions about the input views. Our key insight is that masks belonging to the same object in different views shall correspond to the same Gaussians in the 3D space. Consequently, these Gaussians should be grouped together and assigned an identical group ID.

**Corresponding Gaussians of Mask.** Based on this intuition, we first associate each 2D segmentation mask with its corresponding 3D Gaussians. Specifically, we splat all 3D Gaussians onto the camera frame, using the camera pose of each input view. Subsequently, for each mask within the image, we identify 3D Gaussians whose centers are projected within that mask. Those Gaussians should be identified as representatives of the mask in 3D and can be used as guidance for associating masks from different views.

Notably, 2D segmentation masks typically describe the shape of foreground objects as observed from the current camera view. However, a significant proportion of Gaussians do not contribute to the pixels in the 2D segmentation mask, as they represent objects in the far back of the 3D scene. In Fig. 4, we show one example in column 3.

To address this challenge, we propose incorporating depth information as guidance to select Gaussians corresponding to foreground objects. We first render the depth map for each view. Given a mask $m$ in that

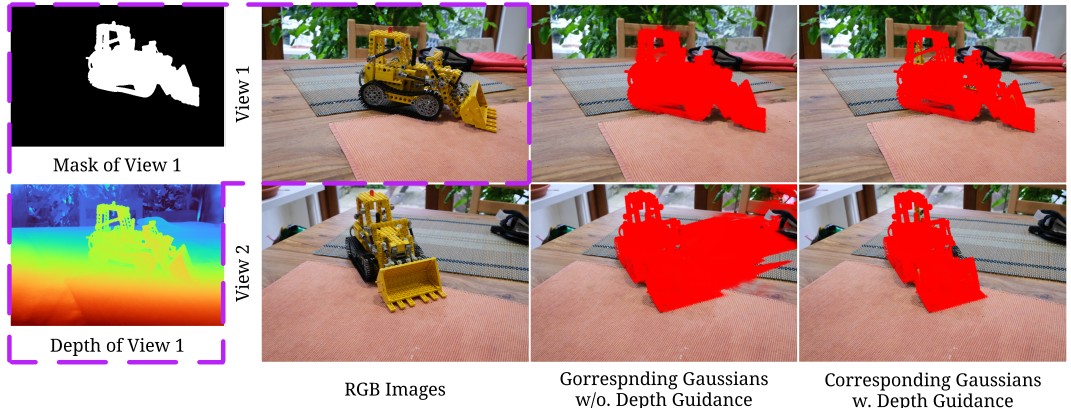

Figure 4: **Corresponding Gaussians with Depth Guidance.** In View 1, we select Gaussians (colored in red) that are splatted inside the mask of the bulldozer. As shown in column 3, some of these Gaussians not only belong to the bulldozer but also represent background objects, as seen in View 2. To refine the selection, we render the depth map and retain only the Gaussians within the specified depth region, ensuring they correspond to the bulldozer's mask. As shown in column 4, the final selection with depth guidance consists primarily of Gaussians belonging to the bulldozer.

view, we extract the corresponding depth values, denoted as $D$, and compute its minimum and maximum depths, $D_{\min}$ and $D_{\max}$, respectively. To mitigate the impact of inaccurate masks, we filter out outlier depths using the following procedure. We compute the first quartile ($Q_1$) and third quartile ($Q_3$) as:

$$Q_1 = \text{Quantile}(D, 0.25), \quad Q_3 = \text{Quantile}(D, 0.75). \tag{2}$$

The interquartile range (IQR) is then defined as $\text{IQR} = Q_3 - Q_1$. To refine the depth range, we define the maximum inlier depth as:

$$D_{\max\_\text{inlier}} = Q_3 + f \cdot \text{IQR}, \tag{3}$$

where $f$ is the outlier factor, set to 1.0 if not explicitly defined. The inliers' depth range is then given by:

$$S = \{z \mid D_{\min} < z < D_{\max\_\text{inlier}}\}. \tag{4}$$

Finally, we select Gaussians that are projected within mask $m$ and lie inside the inlier depth range $S$ as the corresponding Gaussians for mask $m$. As shown in Fig. 4 (column 4), this approach produces a more accurate representation of the foreground objects, as it takes into account the spatial distribution of the Gaussians relative to the camera pose. By leveraging depth information, we can effectively filter out Gaussians that do not contribute to the foreground objects, thereby improving the accuracy of our 3D segmentation. An ablation study demonstrating the significance of depth guidance is presented in Sec. 4.7.

**3D-aware Memory Bank.** Next, to collect and categorize 3D Gaussians into groups and use them to associate masks across different views, we introduce a 3D-aware Memory Bank (see Fig. 3). Given a set of images, we initialize the 3D-aware Memory Bank by storing the corresponding Gaussians of each category in the first image into an individual group and labeling each mask with a group ID the same as its mask label. For each 2D mask of the subsequent view, we first determine its corresponding Gaussians as outlined above. We then either assign these Gaussians to an existing group within the memory bank or establish a new one if they do not share similarities with existing categories in the memory bank. In the following, we elaborate on the details of this assignment process.

**Group ID Assignment via Gaussian Overlap.** To assign each mask a group ID, we aim to find if the current mask has a significant amount of overlapped Gaussians with any groups in the memory bank. We define the similarity between two sets of 3D Gaussians based on their shared 3D Gaussian ratio. Specifically, given the 3D Gaussians corresponding to a 2D mask $m$ (denoted as $\mathcal{G}(m)$ as described above) and the Gaussians of category $i$ (denoted as $\mathcal{G}_i$) in the memory bank, we identify their shared Gaussians as $\mathcal{G}(m) \cap \mathcal{G}_i$

(i.e., Gaussians of the same indices), we then compute the overlap as the ratio of the number of shared Gaussians to number of all Gaussians within mask $m$:

$$Overlap(m, i) = \frac{\#(\mathcal{G}(m) \cap \mathcal{G}_i)}{\#\mathcal{G}(m)}. \tag{5}$$

Note that we do not use the IoU of $\mathcal{G}(m)$ and $\mathcal{G}_i$ for the following reason: as more frames are processed, more Gaussians will be added to the memory bank, which means that $\mathcal{G}_i$ will become larger. Consequently, the IoU threshold needs to be adjusted frequently. In contrast, our overlapping formulation in Eq. 5 is independent of Gaussian number in the memory bank, avoiding manual threshold adjustment. In practice, we set the overlap threshold for declaring a new group ID as 0.1.

Suppose category $i$ has the highest overlap with mask $m$ among all categories in the memory bank, and this overlap value is above a threshold. In this case, we assign the group ID of mask $m$ as $i$ and add the non-overlapped Gaussians in the $i_{th}$ category by $\mathcal{G}_i = \mathcal{G}_i \cup \mathcal{G}(m)$. We establish a new group ID $j$ if none of the existing groups contains an overlap with mask $m$ above the overlapping threshold. We add $\mathcal{G}(m)$ into this new category in the Gaussian memory bank and assign mask $m$ with the new group ID $j$. Note that we ensure each Gaussian is added to only one group in the memory bank by tracking all the Gaussian indices already in the memory bank.

### 3.3   3D Segmentation and Applications.

After the group ID assigning process, masks projected by the same group of Gaussians are supposed to have the same group ID across different views. We splat the group IDs of 3D Gaussians in the memory bank to obtain pseudo ground truth segmentation masks, which are used to learn a 16-dimensional feature vector (*i.e.*, identity encoding (Ye et al., 2024)) for each Gaussian. Identity encoding ($e_i$) (Ye et al., 2024) aims to assign a universal label to each 3D Gaussian for 3D scene segmentation. It can be decoded to a segmentation mask ID through a combination of linear and SoftMax layers. The segmentation label of a pixel (denoted as $m_{x,y}$) is computed by:

$$m_{x,y} = \arg\max\{L(\sum_{i \in N} e_i \alpha_i \prod_{j=1}^{i-1}(1 - \alpha_j))\}. \tag{6}$$

The learning process for identity encoding involves optimizing the feature vectors by comparing the rendered masks with the ground truth 2D segmentation masks. Specifically, we initialize the identity encoding to all zeros for each 3D Gaussian. At each training iteration, we splat the identity encoding given a camera pose to formulate a 2D feature map. A linear layer followed by a SoftMax function is employed to predict a 2D segmentation map given the rendered feature map. Finally, we compute the cross-entropy loss between the predicted segmentation maps and those obtained by rendering Gaussians in the memory bank, as discussed.

After training, our segmentation-aware 3D Gaussians can be readily used for various downstream applications. For instance, we can render segmentation masks of novel views with consistent semantic labels for the same object across different camera poses. Furthermore, Gaussians can also be selected using their identity encoding and manipulated for 3D scene editing tasks, including removal, color-changing, position translation, etc., as will be demonstrated in Sec. 4.6 and appendix.

## 4   Experiments

### 4.1   Experimental Setup

**Datasets.** For quantitative comparison, we use a scene understanding dataset LERF-Mask (Ye et al., 2024), along with two indoor scene datasets: Replica (Straub et al., 2019) and ScanNet (Dai et al., 2017). Additionally, we showcase the robustness of *Gaga* against variations in training image quantity by sparsely sampling the Replica dataset. We present visual comparison results on the commonly used scene reconstruction dataset, MipNeRF 360 (Barron et al., 2021). The quantitative and qualitative evaluations are conducted on the test set, i.e., novel view synthesis results. Details about datasets can be found in the appendix.

**Evaluation Metrics.** Similarly to prior work (Ye et al., 2024), mIoU and boundary IoU (mBIoU) are used to evaluate the LERF-Mask dataset. To evaluate the multi-view consistency of 3D segmentation masks, we

Table 1: **3D query on LERF-Mask.** *Gaga* outperforms previous approaches, showcasing favorable performance in mIoU and BIoU.

| Model | figurines | | ramen | | teatime | |
| --- | --- | --- | --- | --- | --- | --- |
| | mIoU | mBIoU | mIoU | mBIoU | mIoU | mBIoU |
| SA3D (Cen et al., 2023b)* | 24.9 | 23.8 | 7.4 | 7.0 | 42.5 | 39.2 |
| LERF (Kerr et al., 2023)* | 33.5 | 30.6 | 28.3 | 14.7 | 49.7 | 42.6 |
| LEGaussians (Shi et al., 2024) | 34.6 | 32.6 | 31.4 | 18.8 | 42.8 | 35.3 |
| LangSplat (Qin et al., 2024) | 61.9 | 60.9 | 61.9 | 54.7 | 59.8 | 52.7 |
| Gaussian Grouping (Ye et al., 2024)* | 69.7 | 67.9 | 77.0 | 68.7 | 71.7 | 66.1 |
| OmniSeg3D (Ying et al., 2024) | 85.0 | 83.7 | **83.6** | 75.5 | 69.8 | 63.8 |
| IGGT (Li et al., 2026) | 17.2 | 12.2 | 5.6 | 0.9 | 21.4 | 18.3 |
| GOI (Qu et al., 2024) | 63.7 | - | 44.5 | - | 52.6 | - |
| ILGS (Jang & Kim, 2025) | 75.9 | 73.8 | 81.2 | **78.8** | **84.3** | **75.5** |
| *Gaga* (Ours) | **92.3** | **90.8** | 72.0 | 63.3 | 71.2 | 68.4 |

\* denotes results reported in (Ye et al., 2024).

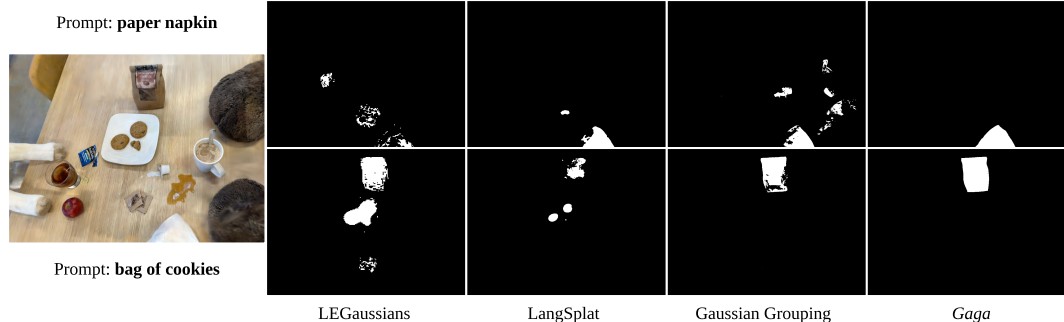

Figure 5: **Qualitative results on LERF-Mask.** Our rendered segmentation exhibits fewer artifacts and delivers more accurate segmentation results than both prior 3D class-agnostic segmentation works and language embedding works.

select eight scenes from the Replica dataset and seven scenes from the ScanNet dataset and use the ground-truth panoptic segmentation for evaluation, disregarding class information. The evaluation process is detailed in the appendix. Note that only Gaussian Grouping and *Gaga* provide 3D mask IDs to render multi-view consistent segmentation masks, we only compare these two methods. To handle differences between predicted and ground truth mask labels, we calculate the best linear assignment based on IoU. Moreover, with IoU = 0.5 as the criterion, we report precision and recall to further evaluate the accuracy of predicted masks. Please note that all numbers in the tables are expressed as percentages.

**Implementation Details.** We use SAM (Kirillov et al., 2023) and EntitySeg (Qi et al., 2023) with the Hornet-L backbone to obtain open-world 2D segmentation. We preprocess the generated raw masks following the method outlined in (Qi et al., 2023), prioritizing those with higher confidence scores by ranking them accordingly. Masks with confidence scores below 0.5 are discarded. For all experiments, we train vanilla Gaussian Splatting (Kerbl et al., 2023) for 30K iterations and train the identity encoding for 10K iterations with all other parameters frozen. We train baseline methods following their official guidance for 40K iterations for fair comparisons.

## 4.2 Open-vocabulary 3D Query on LERF-Mask

We compare our method with nine state-of-the-art methods on 3D scene understanding: LERF (Kerr et al., 2023), LEGaussians (Shi et al., 2024), LangSplat (Qin et al., 2024), SA3D (Cen et al., 2023b), Gaussian Grouping (Ye et al., 2024), OmniSeg3D (Ying et al., 2024), IGGT (Li et al., 2026), GOI (Qu et al., 2024) and ILGS (Jang & Kim, 2025). We note that IGGT is a feed-forward method, and therefore has a clear advantage in inference speed. However, it can utilize at most 50 images per scene. The first three methods and GOI focus on CLIP feature embedding. We calculate the relevancy between rendered CLIP features and query text features following their official implementation. Other methods leverage 2D segmentation masks predicted by SAM (Kirillov et al., 2023). Since SAM does not support language prompts, SA3D,

Table 2: **Quantitative results on Replica and ScanNet.** *Gaga* performs well with both 2D segmentation methods on two datasets, while the performance of Gaussian Grouping varies significantly with different segmentation methods, whereas *Gaga* consistently delivers stable performance.

| Model | 2D Seg. Method | Replica | | | ScanNet | | |
|---|---|---|---|---|---|---|---|
| | | IoU | Precision | Recall | IoU | Precision | Recall |
| Gaussian Grouping (Ye et al., 2024) | EntitySeg | 35.90 | 14.07 | 31.57 | 39.54 | 6.88 | 36.56 |
| *Gaga* (Ours) | | **41.45** | **59.74** | **45.59** | **43.08** | **35.46** | **49.73** |
| Gaussian Grouping (Ye et al., 2024) | SAM | 21.76 | 25.00 | 19.72 | 34.24 | 18.70 | 32.61 |
| *Gaga* (Ours) | | **46.21** | **39.53** | **50.62** | **45.06** | **22.88** | **51.02** |

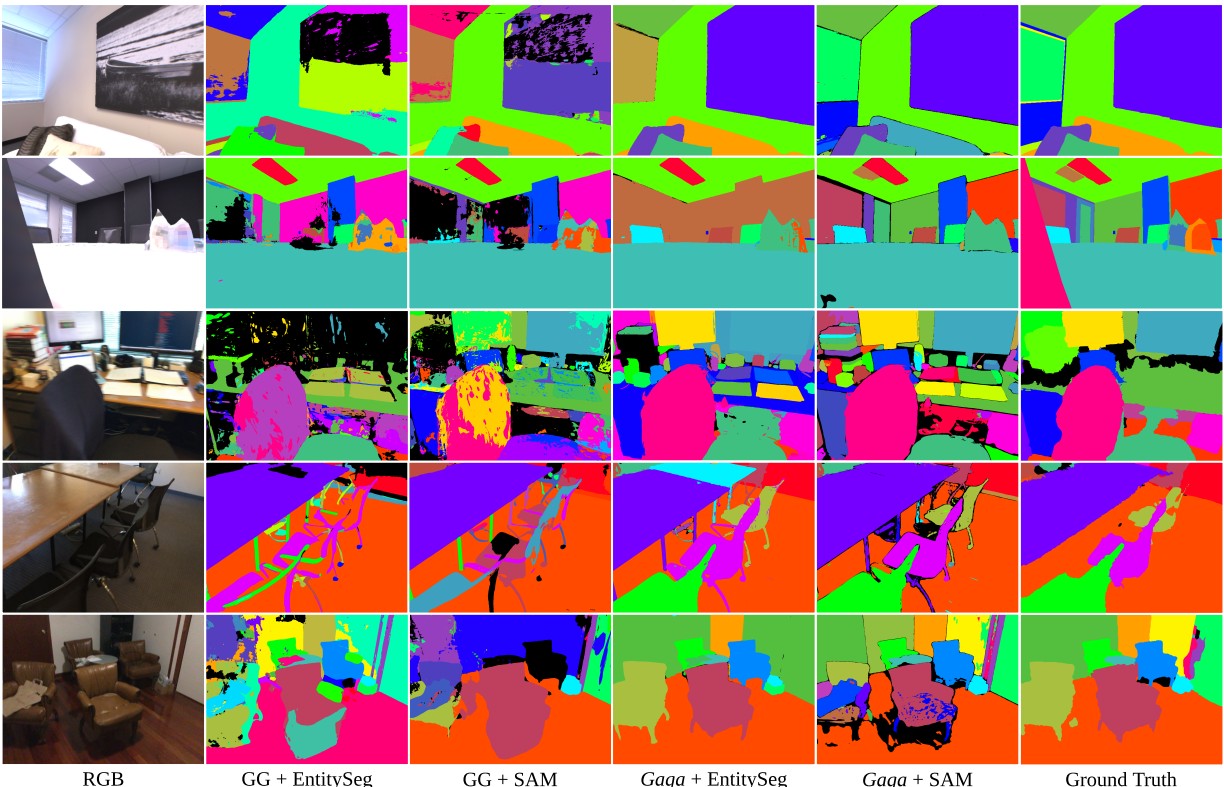

RGB      GG + EntitySeg      GG + SAM      *Gaga* + EntitySeg      *Gaga* + SAM      Ground Truth

Figure 6: **Qualitative results on Replica and ScanNet.** *Gaga* consistently predicts more accurate segmentation masks. Gaussian Grouping often covers the same object with different masks (row 1, 3, 5), creating large empty regions (row 1, 2, 3), or misidentifying similar instances (row 4, 5).

Gaussian Grouping and *Gaga* adopt Grounding DINO (Liu et al., 2023b) to identify the mask ID in a 2D image and pick the corresponding 3D mask. OmniSeg3D does not provide mask IDs, so we use Grounding DINO to identify a pixel that semantically aligns with the text prompt and select the corresponding 3D mask based on a similarity threshold of 0.9. Tab. 1 illustrates that *Gaga* achieves superior results in mIoU and mBIoU compared to previous methods utilizing SAM as 2D segmentation method. As shown in Fig. 5, the visualization results of 3D query tasks with prompts "paper napkin" and "bag of cookies" further demonstrate the advancement of *Gaga*, as *Gaga* provides clearer segmentation masks without artifacts.

## 4.3 3D Segmentation on Replica and ScanNet

Tab. 2 reports results on Replica (Straub et al., 2019) and ScanNet (Dai et al., 2017). Across standard segmentation metrics and for all tested 2D mask providers, *Gaga* performs better on both datasets. This indicates that the gains come from the 3D grouping in *Gaga* rather than the choice of 2D segmentation. The trend is consistent across scenes with different layouts and clutter levels, showing stable performance under

Table 3: **Results on Replica with limited data.** *Gaga* consistently outperforms Gaussian Grouping with both 2D segmentation methods. The percentage of IoU drop shows that *Gaga* has greater robustness against reductions in training data.

| Model | 2D Seg. Method Training Data | EntitySeg IoU ↑ | IoU Drop ↓ | SAM IoU ↑ | IoU Drop ↓ |
|---|---|---|---|---|---|
| Gaussian Grouping (Ye et al., 2024) | 30% | 28.42 | 20.85 | 17.02 | 21.78 |
| *Gaga* (Ours) | | **40.40** | **6.22** | **42.53** | **5.61** |
| Gaussian Grouping (Ye et al., 2024) | 20% | 24.56 | 31.35 | 16.02 | 26.38 |
| *Gaga* (Ours) | | **39.00** | **9.46** | **42.14** | **6.47** |
| Gaussian Grouping (Ye et al., 2024) | 10% | 20.62 | 42.56 | 13.97 | 35.78 |
| *Gaga* (Ours) | | **35.92** | **16.60** | **39.06** | **13.30** |
| Gaussian Grouping (Ye et al., 2024) | 5% | 10.00 | 72.15 | 6.77 | 68.87 |
| *Gaga* (Ours) | | **29.67** | **31.11** | **31.87** | **29.27** |

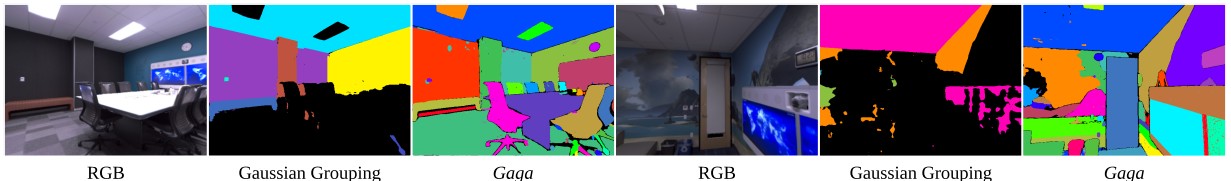

RGB     Gaussian Grouping     *Gaga*     RGB     Gaussian Grouping     *Gaga*

Figure 7: **Qualitative results on Replica with limited data.** The visualizations depict samples when using only 5% of the training data, where *Gaga* still produces high-quality segmentation. In contrast, Gaussian Grouping struggles to track objects accurately and leaves significant empty regions.

domain changes between Replica and ScanNet. Qualitative results appear in Fig. 6. Rows 1–2 show Replica and rows 3–5 show ScanNet. Gaussian Grouping (Ye et al., 2024) often assigns different mask IDs to the same object across views, which leads to identity switches, fragmented masks, and empty regions. In row 4, it fails to separate similar objects. *Gaga* keeps a single, consistent ID per object and uses 3D cues to split adjacent items, so masks are complete and align across views.

### 4.4 3D Segmentation with Limited Data on Replica

To demonstrate the robustness of *Gaga* against changes in training image quantity, we sparsely sample the Replica training set with different ratios. Depicted in Tab. 3, *Gaga* consistently exhibits superior performance in terms of IoU, with approximately a 15% advantage using EntitySeg and a 25% advantage using SAM. Remarkably, when utilizing SAM, *Gaga* surpasses fully trained Gaussian Grouping with just 5% of the training data by more than 10% (31.87% *vs.* 21.76%). We also compute the IoU drop compared to using all training images by:

$$IoU\ Drop(x\%) = \frac{IoU(100\%) - IoU(x\%)}{IoU(100\%)}, \quad (7)$$

where $IoU(x\%)$ denotes the IoU achieved when $x\%$ of the training data is used. *Gaga* exhibits less sensitivity to decreases in the number of training images, as evidenced by smaller values in IoU drop. Visual results are shown in Fig. 7. With just 5% of the training data, *Gaga* delivers accurate segmentation masks, whereas Gaussian Grouping fails to provide masks for a significant portion of objects due to inaccurate tracking.

### 4.5 3D Segmentation on MipNeRF 360

We further evaluate *Gaga* on the diverse MipNeRF-360 dataset (Barron et al., 2021), compared against Gaussian Grouping (Ye et al., 2024). We use two 2D mask providers, SAM (Kirillov et al., 2023) (rows 1–2) and EntitySeg (Qi et al., 2023) (rows 3–4), and visualize two distinct viewpoints to probe multi-view consistency (Fig. 8). Across all settings, *Gaga* yields finer, more contiguous segmentations with sharper boundaries and substantially fewer voids, whereas Gaussian Grouping frequently produces fragmented masks with noticeable artifacts and empty regions. Moreover, the identity association of instances across views is markedly more stable with *Gaga*; in contrast, Gaussian Grouping exhibits pronounced cross-view inconsistencies in rows 1 and 3, manifesting as label flips, missing parts, and spurious regions. These qualitative trends hold for both

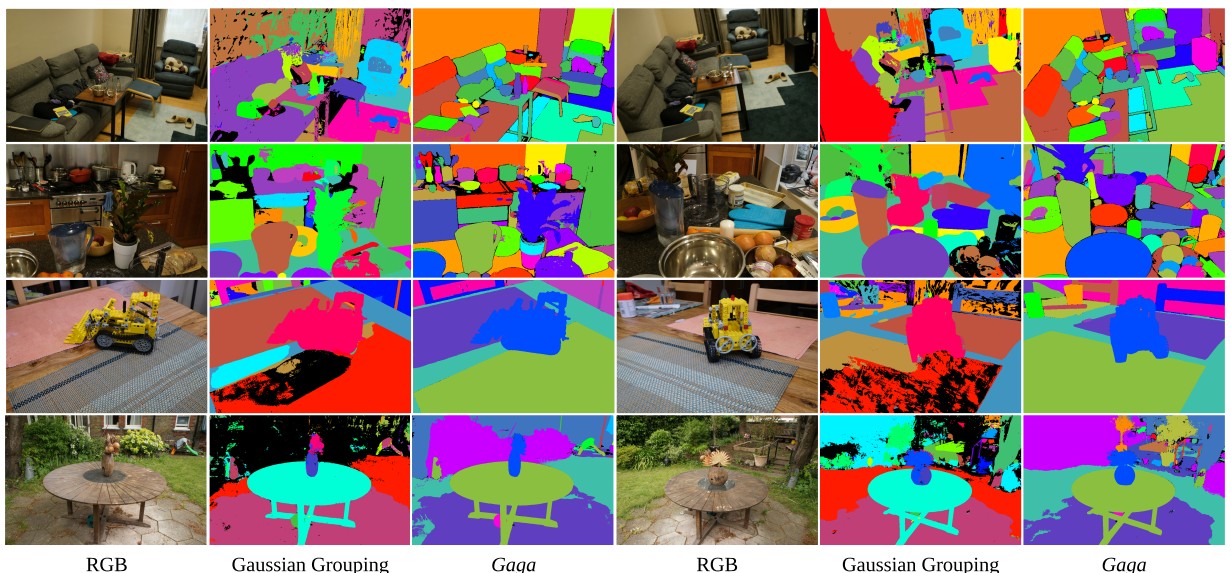

RGB      Gaussian Grouping      *Gaga*      RGB      Gaussian Grouping      *Gaga*

Figure 8: **Qualitative results on MipNeRF 360.** *Gaga* provides superior masks with finer details (row 1 and 2), fewer artifacts and empty regions (row 1, 3 and 4), more consistent object masks across multi-views (wall in row 1, tablecloth in row 3).

Task: Change the cushion's color of   to maroon, remove    Task: Move   closer to the window

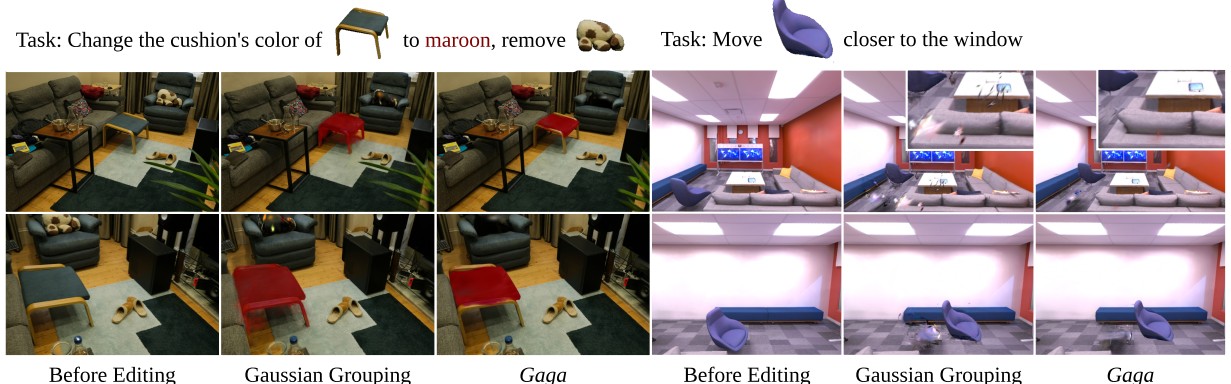

Before Editing      Gaussian Grouping      *Gaga*      Before Editing      Gaussian Grouping      *Gaga*

Figure 9: **Scene manipulation results.** *Gaga* accurately identifies the cushion part of the footstool, whereas Gaussian Grouping colors it entirely. For object removal and translation tasks, *Gaga* generates more precise 3D entities with fewer artifacts, resulting in better visual performance.

SAM and EntitySeg, underscoring that the improvements stem from the 3D reasoning in *Gaga* rather than the choice of 2D mask generator.

## 4.6 Application: Scene Manipulation

*Gaga* achieves high-quality, multi-view consistent 3D segmentation that is directly useful for scene editing. We show results on MipNeRF 360 (Barron et al., 2021) and Replica (Straub et al., 2019) in Fig. 9. We edit color, remove objects, and shift object positions. In the first example, we change the color of the footstool cushion. *Gaga* selects only the 3D Gaussians that belong to the cushion, so the recolor is clean. Gaussian Grouping selects a larger region and turns the whole footstool maroon, and it also spills onto part of the sofa and produces stray floating Gaussians. In the second example, we remove a stuffed animal from the armchair. *Gaga* groups the full object and deletes it with few artifacts. Gaussian Grouping leaves many floating Gaussians in the air and around the sofa. A third example shifts a chair to a new location. Masks from *Gaga* remain stable across views, so the moved chair stays intact and does not leave fragments behind. Gaussian Grouping again shows broken parts and extra floaters. These edits show that accurate and consistent 3D grouping from *Gaga* improves common scene manipulation tasks.

Table 4: **Ablation study.** Our 3D-aware memory bank surpasses the previous baselines on all metrics.

| Baseline | IoU | Precision | Recall |
|---|---|---|---|
| SAM (Upper Bound) (Kirillov et al., 2023) | 60.89 | 57.07 | 67.16 |
| Linear Assignment (Siddiqui et al., 2023) | 1.92 | 1.54 | 0.58 |
| SAM2 (Ravi et al., 2024) | 11.75 | 1.45 | 1.43 |
| w/o. Mask Association | 8.81 | 3.19 | 2.16 |
| Video Tracker (Ye et al., 2024) | 21.76 | 25.00 | 19.72 |
| Our Memory Bank (All Gaussians) | 42.30 | **40.48** | 46.51 |
| Our Memory Bank | 46.21 | 39.53 | 50.62 |
| Our Memory Bank (w. SAM2 (Ravi et al., 2024)) | **49.05** | 30.46 | **53.37** |

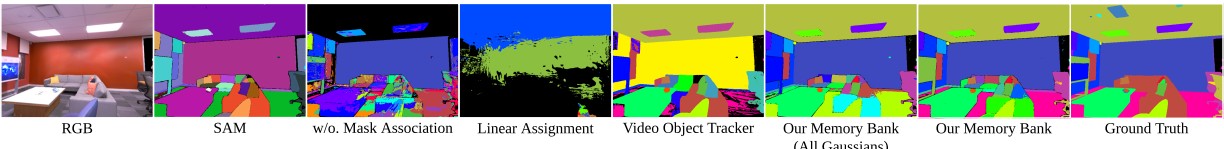

RGB  SAM  w/o. Mask Association  Linear Assignment  Video Object Tracker  Our Memory Bank (All Gaussians)  Our Memory Bank  Ground Truth

Figure 10: **Visual comparison of mask association methods.** *Gaga* with 3D-aware memory bank achieves a superior visual quality and is closer to the ground truth. Notice that the Video Tracker baseline mislabels the wall and floor. Our Memory Bank (All Gaussians) mislabels the floor.

### 4.7 Ablation Study on Mask Association Method

We conduct ablation studies to evaluate the effectiveness of the proposed mask association method on the Replica dataset. The baselines include:

- *Linear Assignment*: Cost-based linear assignment mask association proposed in Panoptic Lifting (Siddiqui et al., 2023).
- *w/o. Mask Association*: Lifting inconsistent 2D masks to 3D.
- *Video Tracker*: Gaussian Grouping (Ye et al., 2024) is employed as a representative method.
- *SAM2*: Use SAM2 (Ravi et al., 2024) as associated masks and lift them to 3D.
- *Our Memory Bank (All Gaussians)*: Selecting all Gaussians splatted to the mask as its corresponding Gaussians.
- *Our Memory Bank*: *i.e.*, *Gaga*.
- *Our Memory Bank (w. SAM2)*: Using SAM2 instead of SAM to get raw 2D masks and associating them with our proposed memory bank.

We also add a baseline *SAM (Upper Bound)*, which uses SAM to process rendered RGBs from Gaussian Splatting and evaluate on each single frame without considering multi-view consistency. This baseline can serve as an upper bound to show the inherent difference between class-agnostic and panoptic segmentation. Results in Tab. 4 and Fig. 10 indicate that *Gaga* with the 3D-aware memory bank achieves superior performance compared to the previous method with video tracker. The cost-based linear assignment method used in (Siddiqui et al., 2023) is not suitable for class-agnostic segmentation tasks, as it involves a significantly larger number of masks compared to panoptic segmentation. Comparison with *Our Memory Bank (All Gaussians)* baseline demonstrates the effectiveness of using depth maps to select corresponding 3D Gaussians for each mask.

## 5 Conclusions

We introduce *Gaga*, a framework that reconstructs and segments open-world 3D scenes using inconsistent 2D masks predicted by zero-shot segmentation models. *Gaga* employs a 3D-aware memory bank to categorize 3D Gaussians and establishes mask association across different views by identifying the overlap between Gaussians projected to each mask. Results on various datasets show that *Gaga* outperforms previous methods with superior segmentation accuracy, multi-view consistency, and reduced artifacts. Additional application in scene manipulation further highlights *Gaga*'s high segmentation accuracy and practical utility.

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

# Appendix

## A  Overview

In this supplementary document, we provide further experimental results, including more results on scene manipulation and sparse view setting in Sec. B. We then delve into more experimental details of the datasets, metrics and implementation in Sec. C. More ablation studies are shown in Sec. D and limitations are discussed in Sec. E.

## B  Supplementary Experimental Results

### B.1  Results on 3D-OVS Dataset

Tab. 5 reports quantitative results on the 3D-OVS dataset (Liu et al., 2023a). We compare *Gaga* with a diverse set of prior methods, including FFD (Kobayashi et al., 2022), LERF (Kerr et al., 2023), 3D-OVS (Liu et al., 2023a), Feature-3DGS (Zhou et al., 2024), Gaussian Grouping (Ye et al., 2024), LEGaussians (Shi et al., 2024), GOI (Qu et al., 2024), LangSplat (Qin et al., 2024), LangSplatV2 (Li et al., 2025), N2F2 (Bhalgat et al., 2024), and OccamLGS (Cheng et al., 2025). Our method achieves a strong 94.5 average score, outperforming most baselines and delivering the best performance on the Bed and Lawn scenes with scores of 97.4 and 97.1, respectively, demonstrating strong open-vocabulary 3D query capability on challenging real-world scenes.

### B.2  Results on ScanNet Dataset

Tab. 6 reports a per-scene comparison between Gaussian Grouping (Ye et al., 2024) and *Gaga* on 14 additional scenes of the ScanNet dataset (Dai et al., 2017). *Gaga* outperforms Gaussian Grouping on most scenes, achieving higher average Mean IoU (41.83% vs. 30.49%), Precision (17.99% vs. 11.02%), and Recall (43.23% vs. 25.35%). This shows the advantage of Gaga for class-agnostic 3D segmentation across diverse scenes.

### B.3  Results on Scene Manipulation

*Gaga* can accurately segment the Gaussians of a 3D object and edit their properties. Using a pre-trained 3D Gaussian model with identity encoding, we employ the classifier trained with identity encoding to predict mask labels for each 3D Gaussian. Subsequently, we select 3D Gaussians sharing the same mask label as the target object and edit their properties for tasks like object coloring, removal, and position translation.

We provide additional results for the downstream scene manipulation task to further demonstrate the prospect of applying *Gaga* to real-world scenarios. On the "counter" scene of the MipNeRF 360 dataset (Barron et al., 2021), we change the color of the flowerpot to cyan and duplicate the glass jar. Gaussian Grouping (Ye et al., 2024) cannot differentiate the plant and flowerpot, whereas *Gaga* generates a more accurate

Table 5: **3D query on 3D-OVS (Liu et al., 2023a).**

| Method | Bed | Bench | Room | Sofa | Lawn | Avg. |
|---|---|---|---|---|---|---|
| FFD (Kobayashi et al., 2022) | 56.6 | 6.1 | 25.1 | 3.7 | 42.9 | 26.9 |
| LERF (Kerr et al., 2023) | 73.5 | 53.2 | 46.6 | 27.0 | 73.7 | 54.8 |
| 3D-OVS (Liu et al., 2023a) | 89.5 | 89.3 | 92.8 | 74.0 | 88.2 | 86.8 |
| Feature-3DGS (Zhou et al., 2024) | 83.5 | 90.7 | - | 86.9 | 93.4 | 88.6 |
| Gaussian Grouping (Ye et al., 2024) | 83.0 | 91.5 | 85.9 | 87.3 | 90.6 | 87.7 |
| LEGaussians (Shi et al., 2024) | 84.9 | 91.1 | 86.0 | 87.8 | 92.5 | 88.5 |
| GOI (Qu et al., 2024) | 89.4 | 92.8 | 91.3 | 85.6 | 94.1 | 90.6 |
| LangSplat (Qin et al., 2024) | 92.5 | 94.2 | 94.1 | 90.0 | 96.1 | 93.4 |
| LangSplatV2 (Li et al., 2025) | 93.0 | 94.9 | 96.1 | 92.3 | 96.6 | 94.6 |
| N2F2 (Bhalgat et al., 2024) | 93.8 | 92.6 | 93.5 | 92.1 | 96.3 | 93.9 |
| OccamLGS (Cheng et al., 2025) | 96.8 | 95.8 | 96.5 | 88.8 | 97.0 | 95.0 |
| *Gaga* (Ours) | 97.4 | 94.4 | 92.5 | 91.0 | 97.1 | 94.5 |

Table 6: **Additional quantitative results on ScanNet**. Comparison between Gaussian Grouping and Gaga across 14 scenes, reported in percentage (%). Scene IDs omit the prefix "scene".

| Method | Metric | 0040_00 | 0101_04 | 0181_01 | 0198_00 | 0217_00 | 0220_001 | 0241_01 | |
|---|---|---|---|---|---|---|---|---|---|
| GG (Ye et al., 2024) | Mean IoU | 28.31 | 34.80 | 18.95 | 32.65 | 33.71 | 30.85 | 29.71 | |
| | Precision | 11.38 | 12.00 | 10.61 | 16.39 | 11.59 | 9.62 | 5.50 | |
| | Recall | 25.33 | 32.61 | 14.58 | 35.71 | 25.00 | 24.39 | 18.18 | |
| *Gaga* (Ours) | Mean IoU | 42.27 | 49.51 | 20.41 | 41.15 | 51.78 | 36.78 | 42.14 | |
| | Precision | 20.88 | 17.65 | 25.00 | 22.00 | 33.90 | 17.00 | 11.21 | |
| | Recall | 50.67 | 58.70 | 18.75 | 39.29 | 62.50 | 41.46 | 39.39 | |
| Method | Metric | 0335_00 | 0564_00 | 0580_00 | 0615_00 | 0616_00 | 0620_00 | 0659_00 | Average |
| GG (Ye et al., 2024) | Mean IoU | 32.76 | 37.79 | 27.40 | 35.65 | 25.94 | 29.71 | 28.64 | 30.49 |
| | Precision | 7.49 | 14.55 | 10.53 | 12.09 | 3.85 | 13.04 | 15.62 | 11.02 |
| | Recall | 23.33 | 36.36 | 24.00 | 31.43 | 9.68 | 23.08 | 31.25 | 25.35 |
| *Gaga* (Ours) | Mean IoU | 46.44 | 41.30 | 37.42 | 45.48 | 32.77 | 48.01 | 50.20 | 41.83 |
| | Precision | 13.78 | 17.65 | 5.61 | 14.78 | 9.09 | 21.88 | 21.43 | 17.99 |
| | Recall | 45.00 | 40.91 | 24.00 | 48.57 | 25.81 | 53.85 | 56.25 | 43.23 |

Task: Change the color of flowerpot to cyan, duplicate the glass jar        Task: Change the color duck to blue, remove the red toy chair

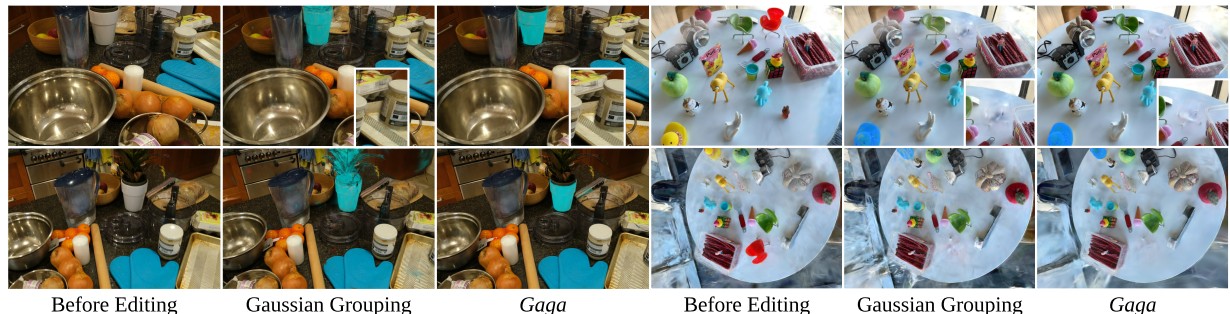

Before Editing        Gaussian Grouping        *Gaga*        Before Editing        Gaussian Grouping        *Gaga*

Figure 11: **Scene manipulation results on MipNeRF 360 and LERF-Mask.** *Gaga* accurately identifies the flowerpot without affecting the color of the plant. Notice that Gaussian Grouping (Ye et al., 2024) creates a cyan region on the wooden door behind. For the object removal and duplication tasks, *Gaga* can also provide more accurate results with fewer artifacts.

segmentation mask. Additionally, *Gaga* produces a clearer boundary and avoids artifacts on the iron tray when duplicating the glass jar.

In the "figurines" scene of the LERF-Mask dataset (Ye et al., 2024), we transform the yellow duck to blue and remove the red toy chair. *Gaga* precisely changes only the duck's color without affecting other objects, and achieves a more thorough removal of the red toy chair.

### B.4 Results on Sparsely Sampled Replica

We provide additional qualitative results for the experiment on the sparsely sampled replica dataset in Fig. 12. As the number of training images decreases, Gaussian Grouping produces more empty regions, *e.g.* the sofa, due to difficulties in accurate tracking under sparse views. *Gaga* exhibits more robust performance under reductions in the number of images.

## C Experimental Details

### C.1 Details on Datasets

We employ the official script from Gaussian Splatting (Kerbl et al., 2023) for colmap to acquire camera poses and the initial point cloud. Consequently, the actual number of images utilized in the experiment might be

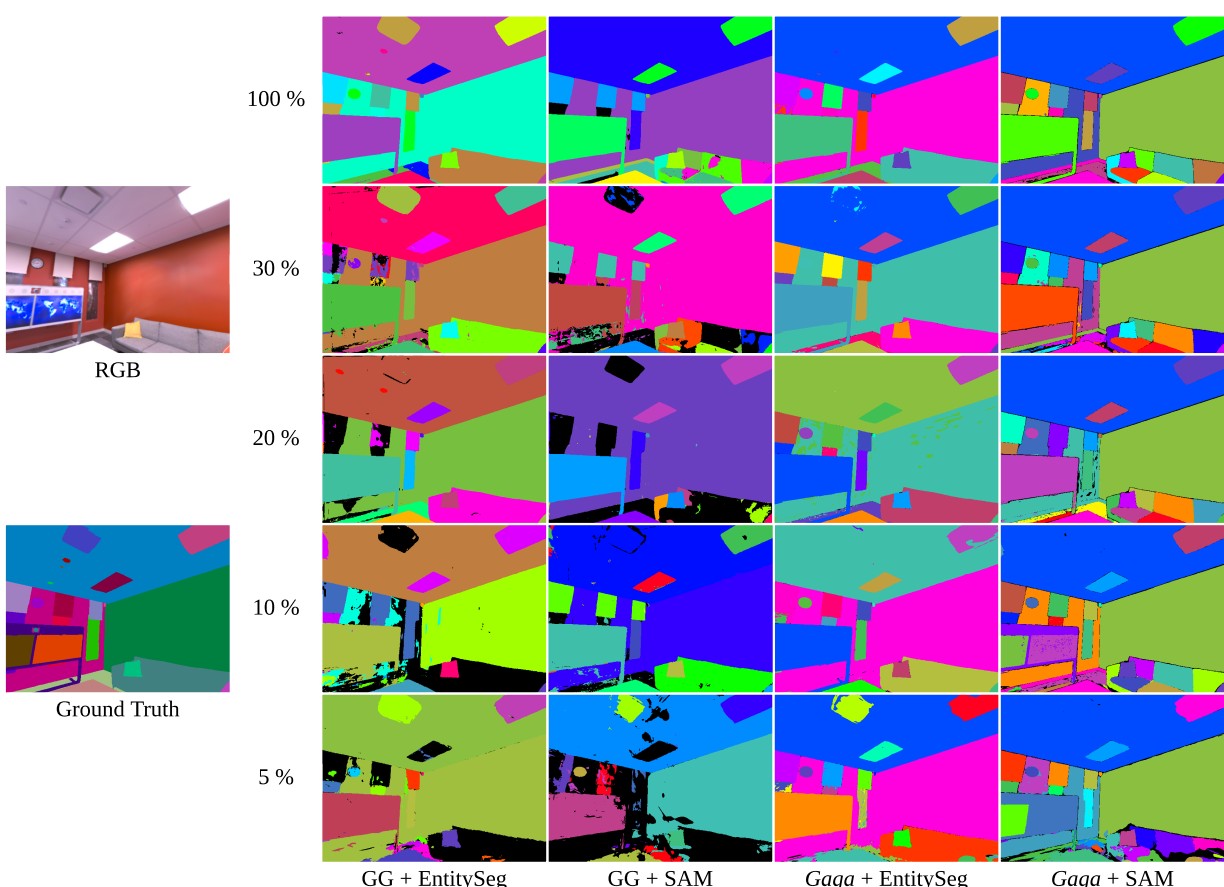

Figure 12: **Qualitative results on the sparsely sampled Replica.** We showcase the novel view synthesis segmentation rendering results provided by Gaussian Grouping and *Gaga* as the percentage of training images employed decreases from 100% to 5%. Gaussian Grouping cannot correctly track the sofa under sparse views and fails to differentiate the ceiling and wall, whereas *Gaga* consistently provides high-quality segmentation.

lower than expected due to colmap process failures. Please refer to Tab. 7 for the scene names used in the Replica and ScanNet datasets.

**LERF-Mask Dataset (Ye et al., 2024).** LERF-Mask is based on the LERF dataset (Kerr et al., 2023) and annotated with tasks and ground truth by the author of (Ye et al., 2024). It contains 3 scenes: figurines, ramen, and teatime. For each scene, 6-10 objects are selected as text queries, and Grounding DINO (Liu et al., 2023b) is utilized to select the mask ID from the rendered segmentation.

**Replica Dataset (Straub et al., 2019).** We select 8 scenes from the entire Replica Dataset the same as (Zhi et al., 2021). We use the rendered results provided by authors of (Zhi et al., 2021) and follow their data processing process: for each scene, we uniformly select 20% images as training data and 20% images as test data from all rendered RGB images. This results in 180 training images and 180 test images for each scene.

**Sparsely Sampled Replica Dataset.** For the same 8 scenes as the previous experiment, we randomly sample 30%, 20%, 10%, and 5% of the total 180 training images, resulting in 54, 36, 18, and 9 training images for each task, respectively. The number of test images remains at 180.

**ScanNet Dataset (Dai et al., 2017).** DM-NeRF (Wang et al., 2023) selects 8 scenes from the entire ScanNet dataset. Each scene has approximately 300 images for training and about 100 images for testing.

Table 7: **Selected scenes in Replica and ScanNet.** We select 8 scenes from the Replica dataset following ([Zhi et al., 2021](#)), and 7 scenes from the ScanNet dataset following ([Wang et al., 2023](#)).

| Dataset | Scene Name | | | |
|---|---|---|---|---|
| Replica | office 0 | office 1 | office 2 | office 3 |
| | office 4 | room 0 | room 1 | room 2 |
| ScanNet | scene 0010_00 | scene 0012_00 | scene 0033_00 | scene 0038_00 |
| | scene 0088_00 | scene 0113_00 | scene 0192_00 | |

Table 8: **Ablation study on the overlap threshold.** If the overlap between the current mask and all groups in the memory bank falls below this threshold, we add this mask to the memory bank as a new group. Results indicate that the default setting of 0.1 generally yields better outcomes.

| Overlap Threshold | figurines | | ramen | | teatime | |
|---|---|---|---|---|---|---|
| | mIoU | mBIoU | mIoU | mBIoU | mIoU | mBIoU |
| 0.01 | 79.6 | 77.5 | 72.0 | 63.1 | 71.2 | 68.2 |
| 0.05 | 91.4 | 89.8 | **72.3** | **63.3** | 70.2 | 67.5 |
| 0.1 | **92.3** | **90.1** | 72.0 | 63.3 | 71.2 | 68.4 |
| 0.2 | 85.8 | 84.3 | 51.3 | 48.7 | **71.7** | **69.4** |
| 0.3 | 85.7 | 84.0 | 45.6 | 43.3 | 71.6 | 69.0 |

We utilize 7 out of the 8 scenes, excluding "scene 0024_00" due to the subpar 3D reconstruction results in both Gaussian Splatting ([Kerbl et al., 2023](#)) and Gaussian Grouping ([Ye et al., 2024](#)).

**MipNeRF 360 Dataset ([Barron et al., 2021](#)).** We downsample the images by a factor of 4, consistent with the setting in ([Ye et al., 2024](#)), to accommodate the large size of the original images. For novel view synthesis evaluation, we set the sample step at 8, the same as the setting in ([Kerbl et al., 2023](#)).

### C.2 Details on Evaluation Metrics

Given the disparate mask label assignments between the ground truth segmentation and the predicted segmentation for 3D objects, we find the best linear assignment between the labels based on IoU for quantitative evaluation. Subsequently, we employ $IoU > 0.5$ as the criterion for precision and recall calculations. We outline the pseudocode for the evaluation procedure in Algorithm 1. Note that all annotated segmentation masks are unavailable during training and are only accessible during evaluation as ground truth.

### C.3 Further Implementation Details

For training vanilla 3D Gaussians, we maintain the same parameter setting as ([Kerbl et al., 2023](#)). To train the identity encoding, we freeze all the other attributes of Gaussians and use the same parameter setting as ([Ye et al., 2024](#)). The identity encoding has 16 dimensions, and the rendered 2D identity encoding is in the shape of $16 \times h \times w$, where $h$ and $w$ represent the height and width of the image. The classifier for predicting mask ID given the 2D identity encoding and selecting Gaussians for editing given the 3D identity encoding shares the same architecture, with 16 input channels. The number of output channels equals the number of groups in the 3D-aware memory bank after associating all images. All experiments are conducted on a single NVIDIA RTX 6000 Ada GPU.

## D Supplementary Ablation Study

### D.1 Ablation Study on Overlap Threshold

We conduct additional ablation studies on the Gaussian overlap threshold using the Replica dataset ([Straub et al., 2019](#)), with SAM ([Kirillov et al., 2023](#)) as the 2D segmentation model. During the group ID assignment process, if none of the existing groups in the memory bank exceed the overlap threshold with the current mask, we add the mask as a new group, indicating the discovery of a new 3D object.

---

**Algorithm 1** Evaluation Metrics

Input *pred_masks* and *gt_masks* are represented in binary format with shape ($n_{image}$, $n_{mask}$, $h$, $w$), where $n_{image}$ is the number of test images, $n_{mask}$ is the number of predicted or ground truth masks, $h$, $w$ are the height and width of test images.

We use `scipy.optimize.linear_sum_assignment` to solve the linear assignment problem.

---

**Function** `evaluate`(pred_masks, gt_masks)

    **Input**: pred_masks (torch.bool), gt_masks (torch.bool)

    **Output**: iou (torch.float), precision (torch.float), recall (torch.float)

  assert len(gt_masks) == len(pred_masks)

  $n_{image} \leftarrow$ len(gt_masks)

  $n_{pred} \leftarrow$ pred_masks.shape[1]

  $n_{gt} \leftarrow$ gt_masks.shape[1]

  iou_matrix $\leftarrow$ torch.zeros(($n_{gt}$, max($n_{gt}$, $n_{pred}$)))

  **for** $i$ **in** $n_{gt}$ **do**

    **for** $j$ **in** $n_{pred}$ **do**

      iou_list $\leftarrow$ []

      **for** $k$ **in** $n_{image}$ **do**

        iou_list.append(`IoU`(gt_masks[$k$][$i$], pred_masks[$k$][$j$]))

      **end for**

      iou_matrix[$i$][$j$] $\leftarrow$ iou_list.mean()

    **end for**

  **end for**

  *gt_indices*, *pred_indices* $\leftarrow$ `linear_assignment`(iou_matrix)

  paired_iou $\leftarrow$ iou_matrix[*gt_indices*][*pred_indices*]

  iou $\leftarrow$ paired_iou.mean()

  $n_{correct} \leftarrow$ torch.sum(paired_iou > 0.5)

  precision $\leftarrow \frac{n_{correct}}{n_{pred}}$

  recall $\leftarrow \frac{n_{correct}}{n_{gt}}$

  **return** iou, precision, recall

---

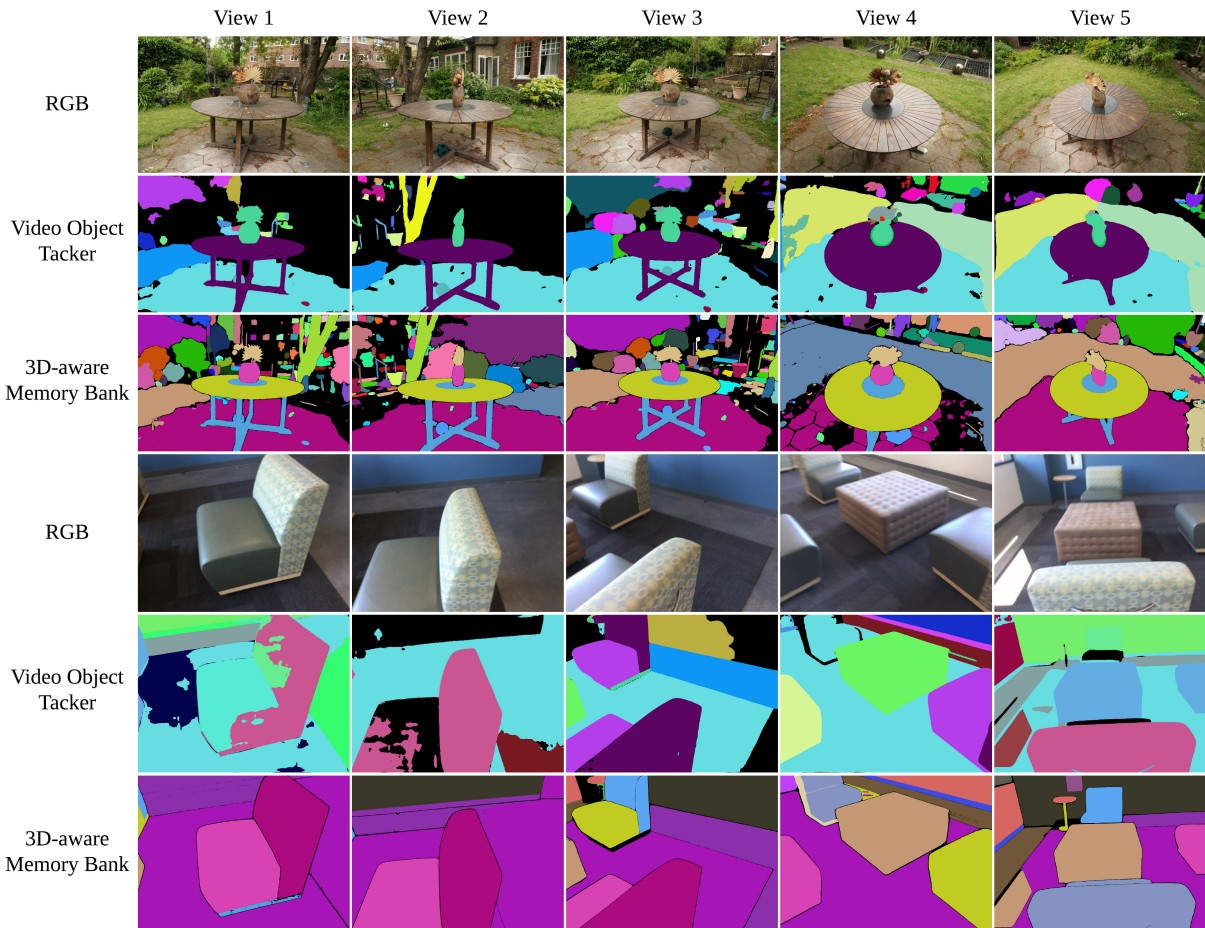

Figure 13: **Visual comparison between different mask association methods.** *Gaga* offers more detailed associated masks, accurately tracks identical objects in the scene and assigns them different mask IDs. Conversely, Gaussian Grouping leaves empty regions in positions where it cannot track masks, and it struggles to provide consistent masks for the same object across views.

Tab. 8 presents our ablation study on the overlap threshold. When set to 0.01, new groups are rarely created, favoring association with existing ones. This yields the highest precision but results in lower IoU performance. Conversely, a threshold of 0.3 leads to frequent creation of new group IDs, achieving the best IoU but significantly reducing precision. To balance performance across all three metrics, we set the threshold to 0.1.

## D.2 Comparison on Mask Association Methods

We present visual comparison results for two mask association methods, video tracker (Cheng et al., 2023a) utilized by (Ye et al., 2024) and *Gaga*'s 3D-aware memory bank, in Fig. 13. In the "garden" scene of the MipNeRF 360 dataset, the video tracker struggles to track objects in the background, whereas *Gaga* provides associated results for each mask. For the scene in the ScanNet dataset, the video tracker fails to distinguish between four identical sofas, resulting in multiple masks for the same object. Additionally, it assigns different mask IDs to the table in two views. In contrast, *Gaga* precisely locates each object, leading to improved mask association results and better pseudo labels for training segmentation features.

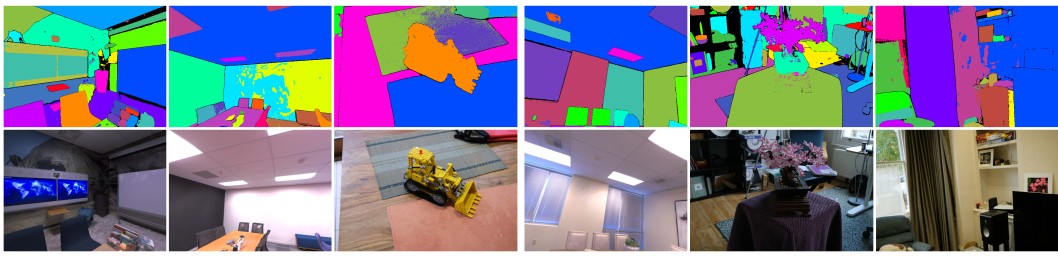

| Over Segmentation | Under Segmentation |

Figure 14: **Examples of failure cases.** We discuss them in two cases: over-segmentation and under-segmentation.

## E    Limitations and Failure Cases

Though *Gaga* achieves SOTA performance compared to existing works, there are a few limitations and future work. First, the optimization process of identity encoding and the rest of the Gaussian parameters are independent, this is because we need to first train 3D Gaussians to acquire their spatial location for mask association. While this pipeline allows for the utilization of any pre-trained 3D Gaussians as input without the need to re-train the entire scene, it does require additional training steps. We aim to enable the joint processing of mask association and identity encoding training in future works.

Secondly, artifacts may occur in the segmentation rendered by *Gaga* due to inherent inconsistency in the 2D segmentation. For example, an object might be depicted as one mask in the initial view but as two separate masks in subsequent views. This ambiguity introduces challenges to our mask association process. Preprocessing steps such as dividing, merging, or reshaping the 2D segmentation masks could potentially resolve this issue and improve grouping results.

We provide examples and analysis of failure cases in Fig. 14. We categorize them into two types: over-segmentation, where the method incorrectly creates separate groups for the same object, and under-segmentation, where distinct objects are incorrectly merged. Over-segmentation mainly arises when the 2D segmentation model splits one object into multiple parts, e.g., the wall in column 1, or when a large object yields limited Gaussian overlap across views, causing multiple masks to be assigned to the same object, e.g., columns 2 and 3. Under-segmentation mainly occurs when the Gaussians associated with one mask mistakenly include objects behind it. A dynamic reassignment mechanism or soft-clustering strategy may help alleviate these issues and is a promising direction for future work.

