# OpenReview forum: "Gaga: Group Any Gausians via 3D-aware Memory Bank"
_TMLR — Accepted by TMLR_

### Review · Reviewer_2rLh · 2026-01-02

**Summary Of Contributions:**

This paper presents Gaga, a 3D-centric framework for open-vocabulary 3D semantic segmentation that resolves the multi-view inconsistency of 2D open-world masks by leveraging geometric consistency in a 3D Gaussian representation.

The core idea is to associate segmentation masks across views based on the overlap of the underlying 3D Gaussians rather than appearance similarity, temporal tracking, or contrastive feature alignment. Each 2D mask is mapped to the set of Gaussians that contribute to it, and a 3D-aware memory bank groups these Gaussians into object-level clusters using a normalized overlap criterion. This produces multi-view consistent group IDs that serve as pseudo-labels for training identity encodings on the Gaussians, enabling coherent semantic segmentation rendering from novel viewpoints.

By grounding mask association in geometry, the method is robust to sparse viewpoints, large camera pose changes, and segmentation noise, and remains agnostic to the choice of the 2D segmentation model. Experiments show that this approach improves segmentation accuracy and consistency compared to prior open-vocabulary and 3D segmentation methods.

## Key strengths:

- Introduces a simple and principled geometry-based mechanism for multi-view mask association.

- Robust to sparse views and large viewpoint changes.

- Model-agnostic with respect to the 2D segmentation backbone.

- Does not rely on temporal continuity, object tracking, or predefined categories.

## Key weaknesses:

- The pipeline is multi-stage and more complex than end-to-end approaches.

- Performance depends on the quality of the initial 3D reconstruction and 2D masks.

- The grouping and overlap criteria involve heuristic design choices that may affect scalability or generalization.

**Audience:**

Yes

**Audience Explanation:**

Yes. The paper would be of interest to a substantial subset of TMLR’s audience, particularly those working on representation learning, 3D scene understanding, multi-view learning, and the integration of foundation models into structured scene representations.

The work addresses a general and recurring problem — multi-view inconsistency and label instability — using a principled geometric mechanism rather than task-specific heuristics. This makes the contribution relevant beyond the specific application of 3D segmentation, as the idea of enforcing semantic consistency through geometric or structural constraints is of broader interest to the machine learning and representation learning community.

Moreover, the method combines ideas from multi-view geometry, memory-based grouping, and self-supervised pseudo-labeling in a way that aligns well with TMLR’s focus on methodological contributions and generalizable principles, rather than purely application-driven results.

**Broader Impact Concerns:**

The proposed method focuses on technical robustness and consistency for open-vocabulary 3D semantic segmentation and does not introduce new application domains that are inherently harmful. However, a few broader impact considerations are worth noting.

First, fine-grained 3D segmentation and manipulation could potentially be used in surveillance or privacy-sensitive scenarios if combined with large-scale scene capture or persistent mapping of real-world environments. Although this is not the intended use, acknowledging this dual-use potential and encouraging responsible deployment would be appropriate.

Second, the method relies on foundation models for 2D segmentation and vision-language alignment, and any biases or systematic errors in these models may propagate into the 3D representation, potentially affecting fairness and reliability across different object types, environments, or cultural contexts.

Third, the computational and environmental cost of large-scale 3D reconstruction and multi-view processing is not discussed and could be briefly acknowledged.

**Claims And Evidence:**

Yes

**Claims Explanation:**

The summary focuses on geometric consistency because it is the paper’s main conceptual contribution. Instead of matching masks using appearance, language features, or temporal continuity, the method associates masks across views based on shared 3D Gaussians, which provides a stable and view-invariant reference.

This design explains the reported robustness to sparse views and large camera pose changes and why the approach is model-agnostic with respect to the 2D segmentation backbone. The 3D-aware memory bank and normalized overlap criterion enable scalable and consistent grouping without manual labels or tracking.

The listed strengths and weaknesses follow from this design: geometry-based association improves robustness and consistency, but the multi-stage pipeline and reliance on reconstruction and mask quality introduce complexity and potential sensitivity.

**Requested Changes:**

1. Clarify failure modes and limitations more explicitly
The paper would benefit from a clearer and more systematic discussion of when and why the proposed geometric grouping can fail, for example in cases of severe reconstruction errors, highly fragmented or inconsistent 2D masks, or heavy occlusion. Making these limitations explicit would improve the paper’s credibility and help readers understand the practical scope of the method.

2. Analyze sensitivity to overlap threshold and grouping hyperparameters (Table 6)
Although the normalized overlap metric is appealing, the method still depends on heuristic thresholds for group creation and assignment. A more thorough sensitivity analysis or principled justification of these hyperparameters would strengthen confidence in the robustness and generalizability of the approach.

3. Report computational cost and scalability more clearly
The memory bank and grouping process may become expensive for large scenes or long sequences. The paper should provide clearer runtime, memory, and scalability analysis, including how performance and cost scale with the number of Gaussians, views, and objects.

4. Compare against stronger or more recent baselines
While the experimental comparison is solid, the paper would be stronger if it included additional comparisons to the most recent or conceptually closest methods, especially those that also aim to enforce multi-view consistency through latent alignment or joint optimization. In particular, a direct comparison with [1] would be valuable, as both methods address multi-view inconsistency but through fundamentally different mechanisms (geometric overlap versus identity-aware latent regularization).

#### [1] Identity-aware Language Gaussian Splatting for Open-vocabulary 3D Semantic Segmentation (ICCV24)

---

> ### Author Response · Authors · 2026-03-25
> **Authors’ Response**
>
> We sincerely thank you for your thoughtful and balanced review. We greatly appreciate your recognition that our geometry-based grouping offers a simple and principled way to address multi-view inconsistency while remaining model-agnostic. We are also grateful for your constructive feedback. We kindly note that the revised manuscript includes additional analyses and discussion marked in red that help address these concerns in the responses below.
>
> **Clarification on failure modes.** We have added examples and analysis of failure cases in Sec. E, *Limitations and Failure Cases*. We discuss two main types of failure. Over-segmentation mainly occurs when a large object has limited Gaussian overlap across views, causing multiple groups to be assigned to the same object. It can also happen when the 2D segmentation model produces masks that are too small. Under-segmentation mainly occurs when the Gaussians associated with one mask incorrectly include objects behind it.
>
> **Sensitivity to the hyperparameter.** The only heuristic threshold in our method is the overlap threshold. We set it to 0.1, as this value provides stable and strong performance across all evaluated datasets. When the threshold is smaller, two Gaussian groups are more easily merged into the same 3D group. In practice, this often introduces errors, since the unprojection from 2D masks to 3D Gaussians can be noisy due to artifacts. This setting tends to favor precision but leads to lower IoU. In contrast, when the threshold is larger, new groups are created more frequently. Since we only select the front-layer Gaussians based on depth, some Gaussians belonging to the same object may be missed in a given view, making it difficult to merge two groups even when they belong to the same object. This setting can improve IoU in some cases but significantly reduces precision. To balance all three metrics, we set the threshold to 0.1.
>
> **Computational cost and scalability.** Our memory bank stores only Gaussian IDs, i.e., one integer per Gaussian, so the additional memory overhead remains minimal even for large scenes or long sequences. We provide runtime and memory analysis on the LERF-Mask dataset in the table below.
>
> | Scene     | Training Time | Peak GPU Memory | Number of Gaussians | Model Size |
> | --------- | ------------- | --------------- | ------------------- | ---------- |
> | Figurines | 91 min        | 32.6 GB         | 2,070,714           | 1.95 GB    |
> | Ramen     | 44 min        | 13.4 GB         | 594,764             | 0.65 GB    |
> | Teatime   | 73 min        | 36.3 GB         | 2,390,604           | 2.26 GB    |
>
> **Stronger and more recent baselines.** We have added comparisons with several stronger and more recent baselines, including IGGT [1], GOI [2], ILGS [3], LangSplatV2 [4], N2F2 [5], and OccamLGS [6], in Tab. 1 and Tab. 5.
>
> ---
>
> [1] Li, Hao, et al. "IGGT: Instance-Grounded Geometry Transformer for Semantic 3D Reconstruction." arXiv preprint arXiv:2510.22706 (2025).
>
> [2] Qu, Yansong, et al. "GOI: Find 3d gaussians of interest with an optimizable open-vocabulary semantic-space hyperplane." Proceedings of the 32nd ACM international conference on multimedia. 2024.
>
> [3] Jang, SungMin, and Wonjun Kim. "Identity-aware language gaussian splatting for open-vocabulary 3d semantic segmentation." Proceedings of the IEEE/CVF International Conference on Computer Vision. 2025.
>
> [4] Li, Wanhua, et al. "LangSplatV2: High-dimensional 3d language gaussian splatting with 450+ fps." arXiv preprint arXiv:2507.07136 (2025).
>
> [5] Bhalgat, Yash, et al. "N2f2: Hierarchical scene understanding with nested neural feature fields." European Conference on Computer Vision. Cham: Springer Nature Switzerland, 2024.
>
> [6] Cheng, Jiahuan, et al. "Occam's LGS: An Efficient Approach for Language Gaussian Splatting." arXiv preprint arXiv:2412.01807 (2024).

---

### Review · Reviewer_ASwj · 2026-01-26

**Summary Of Contributions:**

This paper presents Gaga, a framework that leverages zero-shot segmentation models to segment open-world 3D scenes represented by 3D Gaussians. Gaga maintains a 3D memory bank that clusters Gaussians by semantic category, enabling consistent alignment of 2D masks across diverse viewpoints. For each view, it lifts a predicted 2D mask into 3D using the camera parameters, then matches it to the memory-bank category whose Gaussians yield the largest spatial overlap with the lifted region. The mask is assigned to an existing category or used to initialize a new one. Gaga further encodes category information, splats the resulting features back onto the image plane, and uses a lightweight linear decoder to predict a 2D segmentation map. During training, this map is supervised by comparing it with the segmentation masks derived from the 3D memory bank. Experiments on Replica, ScanNet, LERF-Mask, and MipNeRF 360 demonstrate the effectiveness of the proposed approach across diverse datasets.

Strengths

1. Novel 3D memory-bank design. The paper proposes a 3D memory bank that groups Gaussians by semantic category, enabling consistent cross-view alignment of 2D masks produced by off-the-shelf zero-shot segmentation models.

2. Broad empirical coverage. The method is evaluated on four datasets (Replica, ScanNet, LERF-Mask, and MipNeRF 360), suggesting applicability across varied scene types and data sources.

3. Robustness to reduced views. The paper demonstrates robustness under sparser training/view sampling on Replica, indicating the approach can tolerate fewer observations during training.

Weaknesses

1. Limited dataset scale per benchmark. Although the paper claims effectiveness across diverse datasets, it uses only 4–7 scenes per dataset. This is a small subset—especially for ScanNet (1000+ scenes)—and may limit the diversity of object categories and scene layouts covered, weakening the generalization claim.

2. Insufficient evaluation for 3D query performance. The 3D query results are reported on only one dataset. To support generality, this evaluation should be shown on at least two datasets.

3. Inaccurate phrasing about pose robustness. The paper describes robustness to camera pose variations, but the presented “sparsely sampled images” setting primarily tests robustness to the number/density of input views, not robustness to pose errors. Pose-robustness would require experiments with noisy camera poses.

4. Missing recent baseline comparisons, such as iGGT (Instance-Grounded Geometry Transformer for Semantic 3D Reconstruction), which could be a strong point of reference for semantic 3D reconstruction.

5. Lack of runtime analysis. The paper does not provide any speed or efficiency evaluation, making it unclear how long the full framework takes—from 3D reconstruction to producing the final segmentation masks—and whether it is practical for real-world use.

**Audience:**

Yes

**Audience Explanation:**

Yes, some readers in TMLR's audience may find the idea of leveraging a 3D memory bank for open-world segmentation with 3D Gaussians interesting. However, the paper currently lacks sufficiently comprehensive experiments to convincingly demonstrate the method’s effectiveness and practical usability. In particular, evaluations are conducted on a very limited number of scenes per dataset, and key claims (e.g., generalization and robustness) are not thoroughly validated across multiple datasets and settings.

**Claims And Evidence:**

Yes

**Claims Explanation:**

The paper provides a detailed experimental setup, including concrete hyperparameter and implementation settings. It also includes pseudocode for metric computation, which improves clarity and makes the results easier to reproduce.

**Requested Changes:**

Increase evaluation scale per dataset. To support the generalization claim, evaluate on a larger and more representative subset of each benchmark (well beyond 4–7 scenes per dataset, especially for ScanNet) and report results with broader category and scene diversity.

Extend 3D query experiments to multiple datasets. Report 3D query performance on at least two datasets to show the conclusions are not dataset-specific.

Clarify and validate the pose-robustness claim. Either revise the wording to describe robustness to sparser view sampling (number of input views), or add experiments with noisy camera poses (ideally with controlled noise levels) to directly test robustness to pose errors.

Add comparisons with recent strong baselines. Include additional state-of-the-art baselines for semantic 3D reconstruction (e.g., iGGT).

Provide runtime/efficiency analysis. Report end-to-end runtime from reconstruction to final segmentation masks (and, if relevant, memory usage), including per-stage breakdown, to assess practical usability.

---

> ### Author Response · Authors · 2026-03-25
> **Authors’ Response**
>
> We sincerely thank you for your careful and constructive review. We greatly appreciate your recognition of the novelty of our 3D memory-bank design, our evaluations across multiple datasets, and the robustness of our method under reduced-view settings. We also thank you for your valuable suggestions for more comprehensive experimental validation. We kindly note that our revised manuscript includes additional experiments and clarifications, marked in red, that help address these points in the responses below.
>
> **Additional scenes on ScanNet.** Our evaluation on Replica with 8 scenes follows prior work [1], and our evaluation on ScanNet with 7 scenes follows prior work [2]. In the revised manuscript, we further provide results on 14 additional ScanNet scenes [3] in Tab. 6, where Gaga consistently outperforms the baselines. Due to the substantial time required for both 3D reconstruction with Gaussian Splatting and 2D mask processing with SAM, evaluating on this scale is common practice in prior work, and substantially larger-scale evaluation is costly.
>
> **Additional 3D query experiments.** We further report 3D query results on the 3D-OVS dataset [4], with comparisons against 11 baselines in Tab. 5. Gaga achieves strong performance on this dataset as well, showing that our conclusions on 3D query are not specific to a single benchmark.
>
> **Clarification on pose robustness.** We do not claim pose robustness as a core contribution of this work. Rather, our claim is that Gaga is more robust in limited-training-image scenarios, where sparsely sampled views often exhibit larger camera pose gaps and less overlap between neighboring images. We have clarified this point in the revised manuscript.
>
> **More recent strong baselines.** We have added comparisons with several more recent baselines, including IGGT [5], GOI [6], ILGS [7], LangSplatV2 [8], N2F2 [9], and OccamLGS [10], in Tab. 1 and Tab. 5.
>
> **Runtime analysis.** While Gaga introduces an additional optimization stage for segmentation-aware Gaussian refinement, the training overhead is small. As shown below, on the LERF-Mask dataset, Gaga takes only about 5 minutes longer per scene on average than Gaussian Grouping.
>
> | Scene | Gaussian Grouping | | | Gaga | | | |
> |--|--|--|--|--|--|--|--|
> |  | Mask Association | GS + Identity Encoding Training (40K Steps) | Total | Mask Association | GS Training (30K Steps) | Identity Encoding Training (10K Steps) | Total |
> | figurines | `20:24` | `58:08` | `1:18:32` | `24:40` | `36:29` | `29:23` | `1:30:32` |
> | ramen     | `12:52` | `34:12` | `47:04`| `07:31` | `19:43` | `16:42` | `43:56` |
> | teatime   | `18:03` | `54:53` | `1:12:56` | `24:53` | `28:14` | `26:01` | `1:19:08` |
> | **Average**   | `17:06` | `49:04` | `1:06:10` | `19:01`           | `28:48` | `24:08` | `1:11:12` |
>
>
>
> | Scene      | Gaussian Grouping (min) | Gaga (min) |
> |------------|--------------------------|------------|
> | Figurines  | 78                       | 91         |
> | Ramen      | 46                       | 44         |
> | Teatime    | 73                       | 79         |
> | **Average**| **65.7**                 | **71.3**   |
>
> ---
>
> [1] Shuaifeng Zhi, Tristan Laidlow, Stefan Leutenegger, and Andrew J. Davison. In-place scene labelling and
> understanding with implicit scene representation. In ICCV, 2021
>
>
> [2] Bing Wang, Lu Chen, and Bo Yang. Dm-nerf: 3d scene geometry decomposition and manipulation from 2d
> images. In ICLR, 2023.
>
> [3] Dai, Angela, et al. "Scannet: Richly-annotated 3d reconstructions of indoor scenes." Proceedings of the IEEE conference on computer vision and pattern recognition. 2017.
>
> [4] Liu, Kunhao, et al. "Weakly supervised 3d open-vocabulary segmentation." Advances in Neural Information Processing Systems 36 (2023): 53433-53456.
>
> [5] Li, Hao, et al. "IGGT: Instance-Grounded Geometry Transformer for Semantic 3D Reconstruction." arXiv preprint arXiv:2510.22706 (2025).
>
> [6] Qu, Yansong, et al. "GOI: Find 3d gaussians of interest with an optimizable open-vocabulary semantic-space hyperplane." Proceedings of the 32nd ACM international conference on multimedia. 2024.
>
> [7] Jang, SungMin, and Wonjun Kim. "Identity-aware language gaussian splatting for open-vocabulary 3d semantic segmentation." Proceedings of the IEEE/CVF International Conference on Computer Vision. 2025.
>
> [8] Li, Wanhua, et al. "LangSplatV2: High-dimensional 3d language gaussian splatting with 450+ fps." arXiv preprint arXiv:2507.07136 (2025).
>
> [9] Bhalgat, Yash, et al. "N2f2: Hierarchical scene understanding with nested neural feature fields." European Conference on Computer Vision. Cham: Springer Nature Switzerland, 2024.
>
> [10] Cheng, Jiahuan, et al. "Occam's LGS: An Efficient Approach for Language Gaussian Splatting." arXiv preprint arXiv:2412.01807 (2024).

---

### Review · Reviewer_tfpc · 2026-03-11

**Summary Of Contributions:**

Gaga’s 3D-aware memory bank represents a fundamental shift in how multi-view consistency is achieved in 3D scene segmentation. Unlike prior works that rely on 2D video trackers (assuming continuous, minimal motion) or contrastive learning (which often fails to assign unique, stable labels), Gaga leverages the inherent 3D spatial information of the scene to resolve label inconsistencies.

The most novel technical contributions include:

1. Robust Mask Association via 3D Overlap
Gaga associates inconsistent 2D masks by projecting them into 3D and calculating their spatial overlap with specific groups of 3D Gaussians. This makes the framework robust to significant camera pose variations and enables high-quality segmentation even with sparsely sampled images (e.g., using only 5% of training data), where video trackers typically fail.

2. Depth-Guided Gaussian Selection
To prevent 2D masks from incorrectly capturing background objects, Gaga introduces an automatic depth-filtering mechanism. It uses statistical quartiles to define an inlier depth range, ensuring that only the foreground “representative” Gaussians are associated with a mask.

3. Stable Similarity Metric (Gaussian Overlap Ratio)
The framework introduces a similarity metric that calculates the ratio of shared Gaussians relative to the size of the current mask. This formulation is independent of the total number of Gaussians in the memory bank, allowing the system to use a fixed threshold (0.1) across diverse scenes without manual adjustment—an important improvement over Intersection over Union (IoU) metrics, which often require frequent tuning as the 3D model grows.

By “lifting” fragmented 2D masks into a unified 3D representation via these spatial cues, Gaga provides multi-view consistent identity encodings. This enables cleaner downstream applications—such as precise object removal and recoloring—with fewer artifacts than existing state-of-the-art (SOTA) methods.

**Audience:**

Yes

**Audience Explanation:**

TMLR readers include researchers working in computer vision, 3D scene understanding, neural rendering, and multi-view learning. This paper contributes to these areas by introducing a 3D-aware memory bank that improves multi-view consistent segmentation in 3D Gaussian Splatting scenes. The proposed techniques—such as 3D overlap–based mask association, depth-guided Gaussian filtering, and a Gaussian Overlap Ratio similarity metric—address known challenges in maintaining stable object identities across views, especially under large camera pose changes or sparse image sampling.

These contributions are relevant to ongoing research on 3D Gaussian Splatting pipelines, instance-level scene segmentation, and view-consistent labeling, which are active topics within the machine learning and vision communities represented in TMLR. Additionally, the framework’s ability to improve downstream tasks like object editing, removal, and recoloring in 3D scenes further broadens its relevance to researchers studying 3D scene editing and neural graphics.

Therefore, while the paper is somewhat specialized, it aligns well with several research directions commonly represented in TMLR, suggesting that a subset of its audience would find the findings interesting and useful.

**Broader Impact Concerns:**

I do not see any concenrs

**Claims And Evidence:**

Yes

**Claims Explanation:**

The claims made in the Gaga submission are supported by comprehensive quantitative data, clear visual comparisons, and detailed ablation studies, providing a convincing case for the framework's effectiveness in open-world 3D scene segmentation

**Requested Changes:**

Heuristic Thresholds
Gaga utilizes fixed hyperparameters, specifically a 0.1 Gaussian overlap threshold for group association and a 1.0 outlier factor (f) for depth-guided selection. While the authors provide empirical evidence supporting these values, they remain hardcoded heuristics that may not generalize well to highly complex scenes with extreme scale variations or dense clutter. This limited flexibility can lead to:

Over-segmentation — falsely creating new groups

Under-segmentation — incorrectly merging distinct objects

depending on the density and structure of the scene.

Permanent Gaussian Assignment
The framework follows a strict rule in which each Gaussian is assigned to only one group in the memory bank by tracking unique Gaussian indices. Because the memory bank is constructed iteratively, these assignments become permanent. As a result, a Gaussian mistakenly assigned to a foreground object group (e.g., a car) in an early or occluded view cannot later be reassigned to the correct group (e.g., the ground).

Without a dynamic re-assignment mechanism or soft-clustering strategy, such early mistakes can propagate throughout the training process, potentially leading to persistent segmentation artifacts in the final 3D representation.

In particular, I think aforementioned are some failure cases and I would like to see more results and analysis on these failure cases.

---

> ### Author Response · Authors · 2026-03-25
> **Authors’ Response**
>
> We sincerely thank you for your clear and constructive review. We truly appreciate your recognition of our key technical contributions. We are also grateful for your insightful comments regarding heuristic thresholds and permanent Gaussian assignment. We kindly note that our revised manuscript includes additional experiments and analysis marked in red that help address these concerns in the responses below.
>
> **Failure cases and analysis.** We have added examples and analysis of failure cases in Sec. E, *Limitations and Failure Cases*. In summary, over-segmentation mainly arises when a large object yields limited Gaussian overlap across views, causing multiple masks to be assigned to the same object. Under-segmentation mainly occurs when the Gaussians associated with one mask mistakenly include objects behind it. A dynamic reassignment mechanism or a soft-clustering strategy may help alleviate these issues, and we believe this is a promising direction for future work.
>
> For cases where a Gaussian is mistakenly assigned to a foreground object group, this issue can indeed occur due to the depth-based mask assignment step. However, after obtaining view-consistent masks, we use them to train an identity encoding feature for each Gaussian across multiple views. As a result, even if some Gaussians are incorrectly assigned at the association stage, their impact on the final segmentation rendering results is reduced through multi-view identity supervision.

---

### Review · Reviewer_VjPw · 2026-03-20

**Summary Of Contributions:**

The authors of Gaga have essentially solved the "lost in translation" problem that happens when applying 2D AI to 3D spaces. Specifically, their contributions are:

- A Novel 3D Segmentation Framework: Gaga successfully bridges 2D zero-shot foundation models (like SAM) with 3D Gaussian Splatting (3DGS) to achieve highly accurate, open-world 3D scene segmentation.

- The 3D-Aware Memory Bank: This is the technical crown jewel of the paper. Instead of trying to match objects across images using 2D pixels or optical flow, they project the 2D masks into the 3D space and calculate physical geometric overlap. This memory bank dynamically groups, updates, and unifies mask IDs across entirely different views.

- Enabling High-Fidelity Scene Manipulation: By assigning stable, learnable "ID Encodings" to individual 3D Gaussians, the framework allows for surgical precision in downstream tasks. Users can remove, recolor, or move objects without dragging background artifacts or leaving "ghostly" floating fragments behind.

Strengths:
- Immunity to Sparse/Drastic View Changes: Unlike previous methods that rely on video object trackers (which require tiny, continuous camera movements), Gaga's reliance on actual 3D spatial overlap means it can accurately track an object even if the camera jumps to a completely different angle.

- Plug-and-Play Versatility: The framework is completely agnostic to the 2D segmentation model. You can plug in SAM, EntitySeg, or potentially any future 2D segmentation model, and Gaga will elevate its outputs into consistent 3D representations.

- Exceptional Multi-view Consistency: It elegantly solves the "label flipping" problem (where an object is considered "Chair A" from the front but "Chair B" from the back). This results in exceptionally clean boundaries when rendering the scene from novel views.

Weakness:
- Bottlenecked by the 2D Prior: Gaga is ultimately a "lifter." If the underlying 2D model (e.g., SAM 1) completely fails to detect a heavily camouflaged, highly reflective, or transparent object in the 2D images, Gaga's memory bank won't have the foundational masks to build a 3D group. Also regarding the ID consistent issue, does any more recent 2D segmentation model (e.g. SAM 2) has already resolve?

- Disjointed Optimization Pipeline: The most prominent weakness is the two-stage training approach where the geometric parameters of the 3D Gaussians (position, scale, spherical harmonics) are optimized entirely independently of the semantic identity encodings.

- Lack of Segmentation-Aware Geometry: Because the geometric and semantic pipelines are disjointed, if the initial geometric reconstruction is flawed or entangled at object boundaries, the semantic lifting process cannot exert any corrective gradient pressure. This can lead to floating structures and boundary bleeding.

- Outdated Baseline Comparisons: The manuscript relies heavily on comparisons with early 2024 methods (like Gaussian Grouping) and largely omits rigorous quantitative and theoretical comparisons with the highly relevant wave of late 2024 and 2025 state-of-the-art frameworks. These newer frameworks explicitly tackle the exact same open-world segmentation and lifting consistency challenges using advanced codebooks and unified feature aggregations.

**Audience:**

Yes

**Audience Explanation:**

The findings presented in this paper hold profound and immediate relevance for several distinct sub-communities within the broader machine learning and computer vision audiences targeted by TMLR. The proposed framework cleanly addresses a critical bottleneck at the intersection of neural rendering and high-level 3D spatial perception. TMLR explicitly values scientifically sound, educational methodology over simply chasing benchmark numbers. This paper provides highly competitive empirical performance while simultaneously offering detailed ablation studies that allow readers to mathematically understand why the spatial lifting method works and where its limitations lie.

**Claims And Evidence:**

Yes

**Claims Explanation:**

The primary claims made in the submission are substantiated by a rigorous, comprehensive, and well-structured empirical experimental design. The authors assert that the Gaga framework achieves superior segmentation accuracy, enforces strict multi-view consistency without temporal constraints, gracefully handles severe reductions in training data, and facilitates high-fidelity downstream scene manipulation.

**Requested Changes:**

- Expanded Comparison with Late-2024/2025 Literature: The specific sub-field of 3DGS segmentation has advanced at an unprecedented pace. The authors heavily compare against early 2024 baselines but completely lack discussion regarding concurrent frameworks like InstanceGaussian [1]. The authors must theoretically address and contrast the geometric efficiency of their memory bank with the learned object-level codebooks and unified feature aggregations of these newer frameworks. If direct tabular comparison is impossible, a highly rigorous qualitative theoretical comparison in the "Related Work" section is unconditionally required.

[1] Li, Haijie, et al. "Instancegaussian: Appearance-semantic joint gaussian representation for 3d instance-level perception." Proceedings of the Computer Vision and Pattern Recognition Conference. 2025.

- Theoretical Discussion on Disjointed Optimization: As identified by concurrent literature, the Gaga framework lacks segmentation-aware optimization within the initial 3D Gaussian representation (geometric and semantic optimizations are fully disjointed). The authors must dramatically expand upon this limitation and need to answer: If a more recent 2D segmentation model like SAM2 has already resolve the ID inconsistent issue, is Gaga's framework still valuable?

---

> ### Author Response · Authors · 2026-03-25
> **Authors’ Response**
>
> We sincerely thank you for your thoughtful and detailed review. We greatly appreciate your recognition of the strengths of our 3D-aware memory bank. We also thank you for your constructive questions and suggestions. We kindly note that our revised manuscript includes additional experiments, analysis, and discussion, marked in red, that help address these points in the responses below.
>
> **Bottlenecked by the 2D prior.** We agree that Gaga associates and lifts masks produced by a 2D segmentation model into 3D, and therefore its upper bound is affected by the quality of the 2D masks. However, our goal is exactly to leverage strong off-the-shelf 2D segmentation models, since directly training a 3D counterpart typically requires large-scale 3D annotations, which are difficult to obtain. As shown in our experiments, and as you also noted in the plug-and-play versatility of our method, Gaga is not limited to SAM [1] and can work with other 2D segmentation models, such as EntitySeg (CropFormer) [2]. In practice, better 3D segmentation can be achieved by improving the 2D masks, either through parameter tuning or by adopting stronger 2D segmentation models.
>
> **Compared with SAM2 [3].** We have added two new experiments in Sec. 4.7: (1) a direct comparison with SAM2-based association results, and (2) applying Gaga’s mask association method to masks generated by SAM2. The results show that SAM2 does not provide convincing performance for 3D segmentation. Similar to Gaussian Grouping [4], SAM2 is designed for video segmentation, which struggles in 3D settings where the input views are sparse and unordered. At the same time, Gaga remains valuable in this setting: our Memory Bank (w. SAM2) baseline significantly improves over the SAM2 baseline, and even achieves better IoU and Recall than our default setting using SAM.
>
> **Two-stage pipeline.** This two-stage design is necessary because we need the 3D Gaussian positions to localize mask projections in 3D and perform cross-view association. While we agree that this may appear to be a limitation, it also provides an important advantage: Gaga can be directly applied to any pre-trained 3D Gaussian scene without retraining, making it flexible and easy to use.
>
> **Segmentation-aware geometry.** Thank you for this insightful suggestion. Jointly optimizing geometry and segmentation is indeed an interesting direction, and we agree it could further improve the integration between 3D reconstruction and segmentation. We have added this point to the discussion as an important direction for future work.
>
> **More recent baselines.** We have added comparisons with several more recent baselines, including IGGT [5], GOI [6], ILGS [7], LangSplatV2 [8], N2F2 [9], and OccamLGS [10], in Tab. 1 and Tab. 5. We have also included a discussion of InstanceGaussian [11] in the related work section.
>
> ---
> [1] Kirillov, Alexander, et al. "Segment anything." Proceedings of the IEEE/CVF international conference on computer vision. 2023.
>
> [2] Qi, Lu, et al. "High-quality entity segmentation." arXiv preprint arXiv:2211.05776 (2022).
>
> [3] Ravi, Nikhila, et al. "SAM 2: Segment anything in images and videos." arXiv preprint arXiv:2408.00714 (2024).
>
> [4] Ye, Mingqiao, et al. "Gaussian grouping: Segment and edit anything in 3d scenes." European conference on computer vision. Cham: Springer Nature Switzerland, 2024.
>
> [5] Li, Hao, et al. "IGGT: Instance-Grounded Geometry Transformer for Semantic 3D Reconstruction." arXiv preprint arXiv:2510.22706 (2025).
>
> [6] Qu, Yansong, et al. "GOI: Find 3d gaussians of interest with an optimizable open-vocabulary semantic-space hyperplane." Proceedings of the 32nd ACM international conference on multimedia. 2024.
>
> [7] Jang, SungMin, and Wonjun Kim. "Identity-aware language gaussian splatting for open-vocabulary 3d semantic segmentation." Proceedings of the IEEE/CVF International Conference on Computer Vision. 2025.
>
> [8] Li, Wanhua, et al. "LangSplatV2: High-dimensional 3d language gaussian splatting with 450+ fps." arXiv preprint arXiv:2507.07136 (2025).
>
> [9] Bhalgat, Yash, et al. "N2f2: Hierarchical scene understanding with nested neural feature fields." European Conference on Computer Vision. Cham: Springer Nature Switzerland, 2024.
>
> [10] Cheng, Jiahuan, et al. "Occam's LGS: An Efficient Approach for Language Gaussian Splatting." arXiv preprint arXiv:2412.01807 (2024).
>
> [11] Li, Haijie, et al. "InstanceGaussian: Appearance-semantic joint gaussian representation for 3d instance-level perception." Proceedings of the Computer Vision and Pattern Recognition Conference. 2025.

---

### Decision · Action_Editor_TKqD · 2026-04-30

**Recommendation:** Accept as is

**Audience:**

Yes

**Audience Explanation:**

Gaga is significantly novel in a number of its design choices, the memory bank, the robust mask association via 3D overlap, the depth-guided Gaussian selection to prevent 2D masks from incorrectly capturing background objects and the stable similarity metric. Although the results fall a bit short and the runtime is very long, those technical nuggets are worth the community to look at and potentially draw from. Hence AE follows the majority of the reviewers to recommend acceptance.

**Claims And Evidence:**

Yes

**Claims Explanation:**

After rebuttal reviewers are convinced that the claims are well-supported.